# Utility of long-read sequencing for All of Us

M. Mahmoud [1,2], Y. Huang[3], K. Garimella[3], P. A. Audano [4], W. Wan[3], N. Prasad[5], R. E. Handsaker [6,7], S. Hall[5], A. Pionzio[5], M. C. Schatz [8], M. E. Talkowski [7,9], E. E. Eichler [10,11], S. E. Levy[12] & F. J. Sedlazeck [1,2,13] ✉

The All of Us (AoU) initiative aims to sequence the genomes of over one million Americans from diverse ethnic backgrounds to improve personalized medical care. In a recent technical pilot, we compare the performance of traditional short-read sequencing with long-read sequencing in a small cohort of samples from the HapMap project and two AoU control samples representing eight datasets. Our analysis reveals substantial differences in the ability of these technologies to accurately sequence complex medically relevant genes, particularly in terms of gene coverage and pathogenic variant identification. We also consider the advantages and challenges of using low coverage sequencing to increase sample numbers in large cohort analysis. Our results show that HiFi reads produce the most accurate results for both small and large variants. Further, we present a cloud-based pipeline to optimize SNV, indel and SV calling at scale for long-reads analysis. These results lead to widespread improvements across AoU.

The All of Us project is a landmark initiative by the National Institutes of Health (NIH) in the United States of America to sequence up to one million people using Illumina short-reads and to genotype up to two million people with array data. It is one of the most extensive efforts ever to obtain clinical-grade sequencing for the masses, with a goal of constructing and defining a diverse health database for genomic and other studies across the USA. Already, this effort has released the first 100,000 Illumina whole genome data sets that characterized participants across many ethnicities for genomic variations, focusing on single nucleotide variants (SNVs), small insertions and deletions <50 bp (indels), and genomic alteration ≥50 bp known as structural variants (SVs). Using these data, the program aims to advance personalized medicine by providing preventive and tailored medical care for individuals from globally diverse backgrounds[1]. The endeavor represents a strategic resource to enable genomic sequencing as a leading method for diagnosis and risk assessment, and thus will lead to new insights and improved care for a wide range of genetic diseases.

Understanding the heritability and genetic origins of human genetic diseases requires the accurate and comprehensive identification of all forms of genetic variation along with a detailed recording of phenotypes[2]. Historically, researchers initially focused on characterizing common genetic variation in the human genome in an effort to understand the genetic basis of common diseases and phenotypes in the worldwide population[3]. Nevertheless, they found that the amount of heritability associated with gene variants is often very low, up to ten times smaller than expected in some cases, which many named "missing heritability"[4]. Several hypotheses have been suggested to explain the missing heritability, including poor exploration of SV[5,6], inaccurate characterization of phenotypes[7], and missing rare variants that have a crucial role in some diseases[4,8,9]. Over the last decade, many efforts have been made to address each of these reasons, although all factors continue to challenge disease and traits association studies[10–12].

One major growth area in recent years has been the continued technological improvements made to identify and study SVs,

[1]Human Genome Sequencing Center, Baylor College of Medicine, Houston, TX, USA. [2]Department of Molecular and Human Genetics, Baylor College of Medicine, Houston, TX, USA. [3]Data Sciences Platform, Broad Institute of MIT and Harvard, Cambridge, MA 02141, USA. [4]The Jackson Laboratory for Genomic Medicine, Farmington, CT 06032, USA. [5]Discovery Life Sciences, Huntsville, AL 35806, USA. [6]Department of Genetics, Harvard Medical School, Boston, MA, USA. [7]Program in Medical and Population Genetics, Broad Institute of MIT and Harvard, Cambridge, MA 02141, USA. [8]Department of Computer Science, Johns Hopkins University, Baltimore, MD, USA. [9]Center for Genomic Medicine, Massachusetts General Hospital, Boston, MA, USA. [10]Department of Genome Sciences, University of Washington School of Medicine, Seattle, WA, USA. [11]Howard Hughes Medical Institute, University of Washington, Seattle, WA, USA. [12]HudsonAlpha Institute for Biotechnology, Huntsville, AL 35806, USA. [13]Department of Computer Science, Rice University, Houston, TX, USA. ✉e-mail: fritz.sedlazeck@bcm.edu

which include: insertions, deletions, inversions, duplications, and other rearrangements. Despite being much rarer than SNVs (~0.01% of SNVs per genome), because of their larger sizes, SVs impact a larger number of base pairs per individual[6,13,14]. While very challenging to resolve with short reads, over the past few years, multiple groups have demonstrated the usefulness of long-reads for identifying these types of events[12,15–18]. These research efforts have uncovered novel sequence elements, found biases in the human reference genomes[19–21], resolved the complete telomere-to-telomere (T2T) human reference genome[22], and could begin to demonstrate the impact of complex alleles across various human diseases[17,23].

Currently, there are two major sequencing technologies that provide long reads. The first commercially available technology is from Pacific Biosciences (PacBio), which started with continuous long reads (CLR) with a high error rate (15%)[24,25] using Single Molecular, Real-Time (SMRT) sequencing. Later, they developed high-fidelity reads (HiFi) with an error rate lower than 1%. HiFi reads are typically 15–20 kbp long with a tightly controlled insert size[26]. Today, PacBio sequencing is commonly performed on either their Sequel II or Revio instruments[27]. PacBio has showcased many advantages of long-reads over the years, for example, by boosting our ability to produce highly continuous de novo assemblies (e.g., T2T)[22] and enabling more comprehensive genomic, transcriptomic and epigenomic benchmarks[20].

The second technology, Oxford Nanopore Technologies (ONT), innovated the space with nanopore sequencing, providing longer reads (up to 4 Mbp)[28]. However, these reads often suffer from a higher sequencing error[29], although new "duplex" sequencing is reducing error rates to nearly HiFi levels. ONT has several instruments that provide a range of sequencing capacity to adapt from whole genome population analysis (ONT PromethION) to regional sequencing (ONT MinION and GridION). Over the past years, ONT also innovated de novo assemblies[30] and demonstrated scalability for SV detection[31]. Thus, both technologies currently proved advantageous for detecting complex alleles compared to short-read sequencing[6]. The rapid advancements in long-read sequencing platforms necessitate continuous monitoring of their performance with respect to multiple variant types (e.g. STR, SV) and regions of the genome (e.g. centromere).

Historically, these technologies have seen limited applications in human genetics due to their higher costs, lower throughput, and lower accuracy. Only a few published studies so far have considered more than ten long read genomes, however, these results begin to illustrate the usefulness of the technology and the potential for scaling these technologies[31–33]. For example, one study utilized long-read sequencing to diagnose 13 individuals, with a solving rate of 41.67% (5/12 patients)[34]. They developed special pipelines and used long reads at 46 to 64x coverage to identify SNVs, small indels, and SVs[35]. Still, it is relatively uncertain what the success and utility of long reads at scale will be. While it's clear that they improve the detection of SVs and other complex variants, it is unclear how they compare to short reads for more clinically relevant loci, which are generally less repetitive, especially within exons. Furthermore, if long reads improve only non-coding variant detection (e.g., repeats far from exons), will this be relevant for clinical research? This might not be easy to answer, since most repeat differences and non-coding variants are more difficult to interpret compared to exonic variations.

Answering this question is particularly challenging since the standard of practice for genomics within medically focused studies often does not consider the entire genome, but rather focuses on several key genes that can be prioritized. Usually, these genes have already been shown to be medically important (i.e., have an established impact on certain diseases), and focusing on these genes often reduces costs and reduces the labor needed to review variants of unknown significance. Multiple gene lists are available depending on the physician, the diseases being studied, and the ability of the

technology at hand to assess these. One of the most commonly used gene lists is the ACMG[36], which encompasses 73 genes in their recent release[36]. Two other recent publications postulated their own catalogs of medically relevant but challenging genes. Mandelker et al. showcase 193 genes that are hard to impossible to characterize using traditional short-read sequencing technology[37]. Thus, they showed the need to use long reads to fully capture these genes (e.g., *SMN1& 2, LPA*). Wagner et al. extended this list further based on a superset of 5000 genes[20]. They identified 386 genes as highly challenging and reported to be medically relevant. These genes represent different challenges like complex polymorphism (e.g., *LPA*), high levels of repeats (*SMN1&2*), and interaction with their pseudogene (e.g., *GBA* vs. *GBAP1*)[20].

In this work, we investigate the utility of long-reads for the All of Us program using a combination of publicly available control samples and long-read sample data collected using a range of tissue types and extraction methods from samples previously used inside All of Us to establish the short-read pipeline. We used the control samples to derive a computational pipeline that can accurately identify SNVs and SVs at scale. To make the work scalable and reproducible, the pipeline is implemented using the Workflow Definition Language (WDL) and hosted in a public GitHub repository (https://github.com/broadinstitute/long-read-pipelines) making it possible to run in large data centers and commercial computing clouds. Furthermore, we compare this pipeline with Illumina whole genome data processed with DRAGEN version 3.4.12 (the FDA approved version for this project), which is the All of Us production short-read pipeline, to assess long-read utility. We do so on both "simple" medically relevant genes (4641) and "challenging" medically relevant genes (386) to evaluate the different sequencing technologies. One critical study characteristic we evaluated is coverage: for a fixed amount of financial resources, lower coverage per sample can potentially expand the number of samples analyzed yet requires careful controls to not inflate errors or missing variation. Overall, our study answers the question of the utility and need for long-read sequencing to identify previously hidden variations that likely have implications on medical phenotypes.

## Results

### Optimizing variant detection in cell lines

We first focused on four established HapMap cell lines to assess the long-read ability to cover and comprehensively identify variants across medically relevant genes. We used the widely studied sample NA24385 (Caucasian male), as well as HG00514 (female Han Chinese), HG00733 (female Puerto Rican), and NA19240 (Yoruba male), for which assembly-based analyses are available[33]. The latter samples (HG00514, HG00733, NA19240) are not as well-curated as the NA24385[20,38,39]. Nevertheless, they provide valuable information, since variant calling tools have generally not been previously optimized across these samples to minimize overfitting.

First, we assessed the genome-wide coverage and read length capabilities across these control samples to quantify the ability of the different sequencing technologies at this basic level. Because of the randomized nature of whole genome sequencing, coverage and read length establish the fundamental limitations of variant identification. For example, in an extreme scenario, if there are zero reads spanning a given region of the genome, clearly no variants will be detected there. Moreover, if variants are detected by only a single read they are generally less trustworthy and are often filtered out. One simple but effective strategy is to require at least two or more supporting reads to identify a variant since it is substantially less likely two reads will have the same error at the same position (e.g., assuming a 1% sequencing error, there is a 0.01% chance of two reads having the same error at the same position at random)[12]. In low coverage situations, however, requiring two reads spanning a position to identify a variant will limit the recall of variants,

especially for heterozygous variants since the haplotype-specific coverage is half of the total coverage. Nevertheless, an idealized analysis assuming a Poisson coverage distribution shows 6× coverage is sufficient to recall over 98% of homozygous variants and 8x is sufficient to recall over 90% of heterozygous variants (Fig. 1a, Methods). One notable exception to this model is capturing insertion variants when the length of the variant approaches or exceeds the length of the read, since the amount of coverage that spans the variant will be proportionally reduced by the length of the insertions (Fig. 1b, Methods). This idealized analysis establishes the fundamental limits for variation detection. In real-world analysis, there are additional considerations, especially non-random sequencing errors, repeats, alignment errors, overdispersion in coverage, and other biases that further complicate variant calling. This requires an empirical approach to measure and resolve that we discuss in later sections of the paper.

The short-read Illumina coverage was between 29.76× (NA19240) and 32.50× (NA24385) (see Methods), and across the long-read technologies (Fig. 2a), we obtained an average coverage of 45.29× for ONT and 35.70× for PacBio HiFi. It is important to highlight that ONT required only 1–2 PromethION flow cells per sample to produce this coverage, while PacBio HiFi required several times more using Sequel IIe (HG00514, HG00733, & NA19240 each required four flow cells, NA24385 six flow cells). ONT also generated longer aligned N50 lengths compared to PacBio HiFi, with an aligned N50 length of 20 kbp compared to an aligned N50 length of 11 kbp (Supplementary Fig. 1) (Fig. 2a).

Next, we assessed the ability of the technologies for the identification of SNVs and indels across these samples using state-of-the-art methods. Starting with Illumina, we utilized the Dragen pipeline (v3.4.12)[40] with the exact specification as AoU, resulting in a high recall of 99.32% and precision of 99.63%, leading to a total F-score of 99.47% across GIAB HG002 benchmark v4.2.1[39] (see Methods).

For the long-read technologies, we evaluated several best practice small variant callers (Longshot[41], Pepper/DeepVariant[42], and Clair3[43]) as well as various combinations of them. Figure 2b shows the indel size

ranges that were detected, and Fig. 2d shows the comparison of SNV and indel density across samples. Importantly, Longshot can only identify substitutions (Supplementary Figs. 2 and 3), which makes it complicated to compare this method to the others, which can also identify short indels. DeepVariant and Clair3 both showed a similar number of base substitutions when applied to data generated by PacBio and ONT sequencers. Specifically, DeepVariant identified 4,335,053 substitutions in the PacBio data and 4,709,454 substitutions in the ONT data. Similarly, Clair3 identified 4,597,773 substitutions in the PacBio data and 4,578,168 substitutions in the ONT data. On the other hand, Longshot resulted in an approximately 1.57-fold increase in the number of substitutions for data generated by ONT, although the excess is almost entirely false positives (see below). Overall, a combination of Clair3 and DeepVariant using PacBio HiFi data achieved the best F-score (99.87%) at this coverage level (as shown in Supplementary Fig. 4). For ONT, the F-score is 98.74% from merging the results from Clair3 and DeepVariant (Supplementary Fig. 5). It is worth noting that the improvement from merging Clair3 and DeepVariant for PacBio is marginal compared to using only DeepVariant or Clair3 individually, with a gain of only 0.01% to 0.04% in precision, respectively. Conversely, for ONT, the merge F-score enhancement is 0.70% (DeepVariant) and 2.50% (Clair3) improved over using the individual methods. Furthermore, we assessed the correctness of genotypes across NA24385 and found that Clair3 achieved higher accuracy than DeepVariant. Given this result, we adjusted our merging strategy to utilize the genotype values (GT flags) from Clair3 when available. With the pipeline optimized for one sample, we extended the analysis to HG00514, HG00733, and NA19240 and benchmarked them against their respective gold-standard variant call sets[44]. Clair3 merged with DeepVariant achieved the best results. Regardless, the overall performance is reduced compared to NA24385 (HG002). The difference in the accuracy of variant calling between HG002 and other samples is attributed to the benchmark set regions for HG002 being filtered to exclude repetitiveness or polymorphic complexity regions, as mentioned in Wagner et al.[38] and Li et al. studies[45]. Additionally, the variants

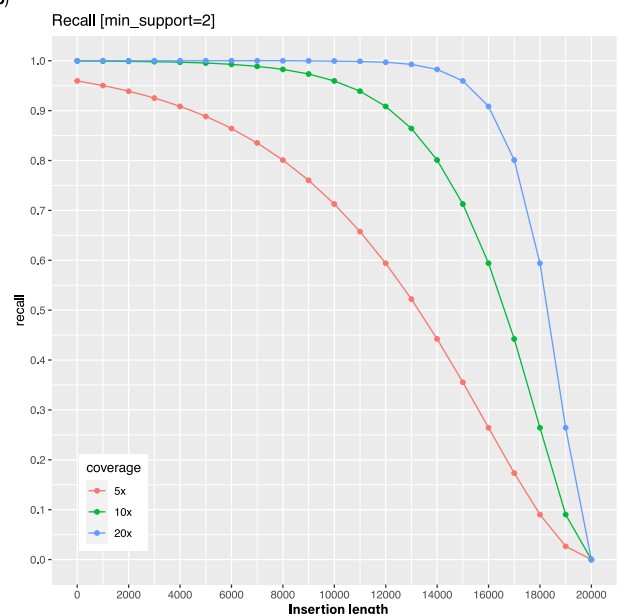

**Fig. 1 | Theoretical assessment of variant calling as a function of coverage and insert size. a** Idealized recall of homozygous and heterozygous variants as a function of overall coverage, assuming at least two spanning reads are required to recall a given variant in the genome. Here, we assume the sequencing coverage follows the Poisson distribution centered on a given overall coverage level, and the coverage will be evenly distributed across maternal and paternal haplotypes following the binomial distribution with $p = 0.5$. **b** Idealized recall of insertion variants of different lengths assuming 20 kbp reads, 5×, 10× or 20× coverage overall, and at least 2 reads spanning the insertion. As plotted, this represents the recall of homozygous variants, although heterozygous variants follow the same distribution as a function of the haplotype specific coverage.

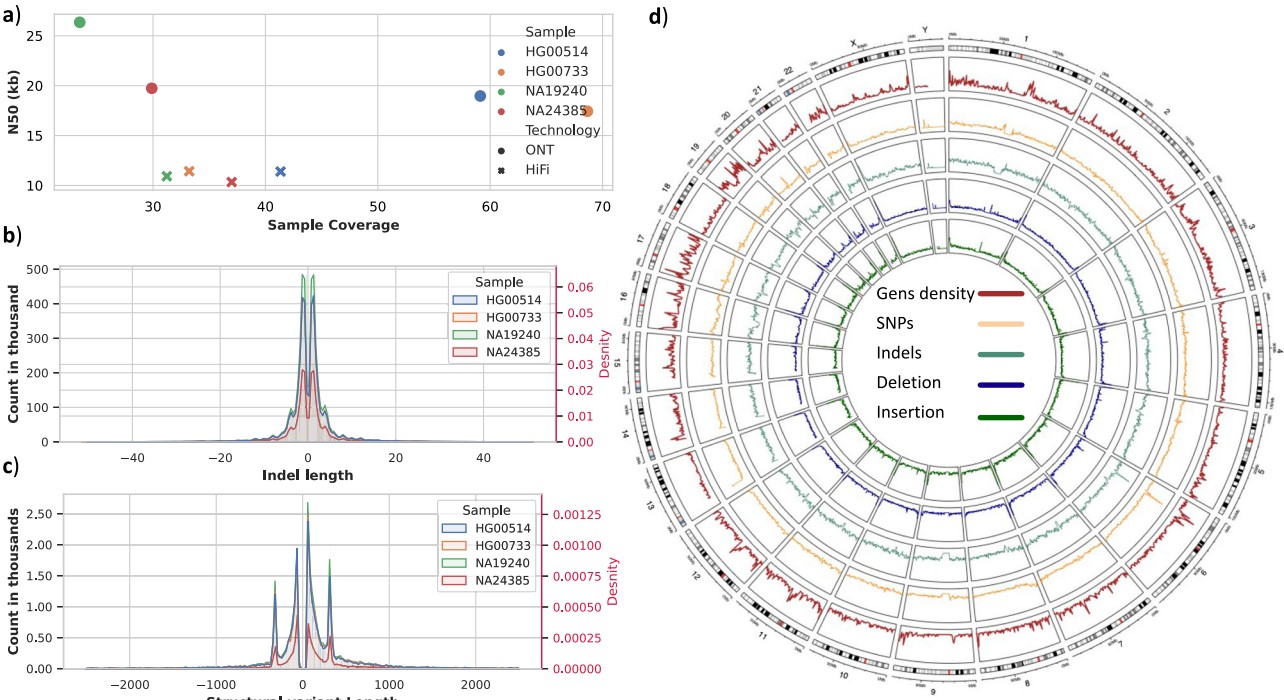

**Fig. 2 | Summary comparison across HapMap samples. a** Coverage on *X*-axis vs N50 on the *Y* axis, for samples NA24385, HG00514, HG00733, and NA19240 (HapMap). **b** Indels distribution for HapMap samples, indels length on the *X*-axis, and count and density (in red) on *Y*-axis (using a merge of DeepVariant and Clair3 for HiFi). **c** SV size distribution for HapMap samples, SVs length on the *X*-axis, and count and density (in red) on the *Y*-axis (using a merge of Sniffles and pbsv HiFi).

**d** Circular plot for HapMap samples from outside in genes density (Brown), SNVs (substitutions) average density between HapMap data (Orange), SNVs (indels) average density between HapMap data (Aquamarine), deletions average density between HapMap data (Blue), insertions average count between HapMap data (Green).

in this sample have been extensively validated, and their accuracy is well-established.

We assessed the technologies' abilities for SV calling in medically relevant genes using Manta starting with NA24385 (see methods). Using the state-of-the-art algorithm Manta developed by Illumina[46], Illumina's F-score is 0.45, due to its inability to identify large insertions. In contrast, we found that a combination of pbsv and Sniffles for ONT and PacBio achieved the best results, with an F-score of 0.93 using HiFi and 0.91 for ONT (see Supplementary Fig. 6). Furthermore, we conducted a titrated coverage comparison for sample HG002, focusing on the two long-read technologies, PacBio and ONT. We evaluated the SV detection F1-score for this comparison (Supplementary Fig. 18). Overall, the trends follow the idealized results presented in Fig. 1, with a sharp increase in accuracy between 5× to 15× coverage and align with other ongoing research[47]. We also observed that below 8× coverage, ONT demonstrated a higher F1 score for SVs compared to PacBio HiFi. However, beyond 8× coverage, there was a significant improvement in the F1-score for PacBio HiFi, exceeding 90%, while the F1-score for ONT remained around approximately 87%. When applied to the other samples, we found that using long-reads improved the accuracy of SV detection, although the F-score was slightly lower at 0.77 (as shown in Supplementary Fig. 7). When comparing the detection of SVs between the samples (as depicted in Fig. 2c), we observed that the frequency of detected SVs was similar.

Overall, the results of this study reproduce several previous findings that long-read sequencing enhances SV calling and achieves a high accuracy of genome-wide SNV and indel calling. We further established an improved variant calling pipeline for SNVs and SVs for long reads that work similarly well for PacBio HiFi as well as ONT and thus makes the comparison across the sequencing technologies faster and easier.

## Performance assessment for All of Us samples across medically relevant genes

To provide a more realistic assessment of the value of long-read sequencing in a clinical research setting, we next utilized long-reads to benchmark the analysis of control samples commercially sourced by the AoU. These samples were sequenced multiple times across the individual AoU genome centers to establish and assess the variability of each center for different tissue sources and preparations. Specifically, we sequenced two anonymized AoU samples T662828295 and T668639440 using ONT, HiFi, and Illumina technologies for different tissue sources (white blood cells and whole blood cells, henceforth WBC and whole blood cells, respectively) and extraction methods (Chemagen and Autogen).

First, when assessing coverage, we observed that in this limited sample set, WBC achieved a better coverage with either Autogen or Chemagen extraction for ONT and HiFi, while the opposite was found for Illumina (whole blood cells achieve a better coverage). The minimum and maximum coverage for AoU samples T662828295 and T668639440 are shown in Supplementary Data S1: Illumina (31.95, 41.68, 34.19, 39.30), HiFi (6.46, 10.50, 5.11, 15.76) and ONT (28.92, 29.26, 28.09, 29.46). This coverage resulted from using one flow cell for ONT and maximum two flow cells for HiFi (Supplementary Fig. 8 and Supplementary Data S1). Furthermore, we observed that the extraction method impacted the N50 alignment length (Supplementary Fig. 9). The average N50 of aligned reads for HiFi (17,612) is greater than that of ONT (15,726 bp). However, ONT produced the longest single read alignment by a large margin (758,354 bp).

For small variant calls (substitutions, insertions, and deletions <50 bp), we assessed the concordance between technologies and found that long- and short-read, agree on approximately 67.55% (median 70.47%). We can explain this reduced concordance level by the lower HiFi coverage and the difficulty in accurately calling

insertions and deletions between technologies. Nevertheless, when we compare substitutions only, the concordance reaches 79.00% on average (Supplementary Data S2). Unsurprisingly, among variants not found by all three technologies, HiFi (5.44%) shows greater SNV concordance with Illumina when compared to ONT (8.46%); except for sample T668639440 whole blood Chemagen, where ONT agreed two folds higher than HiFi with Illumina 15.49% and 6.46%, respectively. This is likely due to the much lower coverage from HiFi (5.11x) compared to ONT (28.09×) data set, as mentioned earlier. Additionally, we noticed an enrichment for Illumina-only identified variants (mean 12.10% and median 8.98%) compared to a lower unique identification on each of the long reads (ONT or HiFi ~2.29%). Of note, the higher Illumina average of 12.10% uniquely identified SNVs was chiefly due to the T668639440 WBC Autogen sample. This outlier substantially skewed the mean, causing 33.54% of variants to be identified uniquely by Illumina. Correspondingly, when we focused only on exon and intron regions (point mutations and indels), the concordance between long- and short-read increased to 81.33% and 77.46%, respectively. Additionally, HiFi showed higher concordance in exons and introns with Illumina (Supplementary Data S3).

We utilized SnpEff[48] across the 24 merged data sets to annotate the merged variants obtained from the three technologies. A total of 10,939,305 variants were processed and annotated. Among them, we identified 7622 variants (0.02%) across the 24 data sets with high-impact annotations, including stop-gained or frameshift variants, which were of particular interest for further analysis.

When we compared the high-impact variants identified by ONT, HiFi, and Illumina, we discovered that the percentage of high-impact variants detected exclusively by Illumina was identical to those detected by both HiFi and ONT, which was just 0.04%. The same percentage was also observed for high-impact variants detected by the union of HiFi and Illumina. However, considering the high-impact variants detected by either ONT or HiFi alone, this percentage increased to 0.05%. Additionally, when evaluating all long-read technologies collectively, the percentage of high-impact variants identified rose to 0.09%, as depicted in Supplementary Fig. 19.

We next analyzed SVs using the three technologies (HiFi, ONT, and Illumina). We identified an average of 24,235 SVs per sample (for each tissue and extraction method), which aligns with previous studies[44]. The percentage of SVs agreed upon by all three technologies is approximately 22.00%; ONT and HiFi agreed on 53.86% of all SVs (31.90% identified only by long read plus 22.00% identified by all reads). However, 22.40% remains that is either detected exclusively by ONT or HiFi, meaning that 32% of the SVs identified by long read were not detected when using short read sequencing. Moreover, approximately 15% of SVs were found exclusively by Illumina.

We studied in depth each of these distinctive variants per technology. We found that 950 SVs were identified uniquely by Illumina in all datasets, and the majority are translocations (68.63%), followed by duplications (11.26%). For ONT, we identified 57 unique SVs, with the majority being deletions (56.14%), followed by insertions (42.11%), and duplications (1.75%). Furthermore, we found that PacBio had the least number of unique SV (54), with the majority being insertions (40.74%), followed by deletions (37.04%), duplications (16.67%), inversions (3.70%), and translocations (1.85%). Based on these results, it is likely that Illumina reports a higher number of false SVs, particularly translocations in healthy individuals, as seen in prior studies[12,32]. Meanwhile, deletions dominated the uniquely-identified SVs from ONT, while insertions dominated for HiFi. Additionally, we utilized vcfanno[49] to annotate the SVs that we identified. We calculated the percentage of SVs that overlap genes (entire gene body) and presented these findings in Supplementary Fig. 19. This might be inflated as larger inversions, for example, might not have a direct impact on genes. Our analysis revealed that the percentage of SVs overlapping genes was lower in Illumina sequencing (~43%) compared to HiFi and ONT sequencing (~54%), as well as all SVs identified by long-read sequencing (~52%). Furthermore, when we filtered out inversions and recalculated the percentages of SVs overlapping genes, we noted a decrease of less than one percentage, as illustrated in Supplementary Fig. 19.

To investigate the clinical utility of long-reads for a program like All of Us, we focused on a set of 4641 genes that are reported to be medically relevant[38]. Most notably, these genes represent non-repetitive or generally non-complex genes of the human genome (see Fig. 3a). The coverage across these genes was similar to the genome-wide coverage across all technologies (see Fig. 3b). In Fig. 3b, we compared normalized gene coverage (average gene coverage divided by sample average coverage) between HiFi and ONT and found larger coverage variability for HiFi likely due to the overall lower coverage.

Similarly, we compared the number of genes with average coverage less than one (henceforth, "uncovered genes") across the sequencing technologies. For Illumina, we identified only three uncovered genes for T668639440 Chemagen (*C4A, C4B*, and *OR2T5*), while Autogen only had two uncovered genes (*C4A* and *C4B*) for the same sample and only one gene for sample T662828295 Autogen (*C4A*). PacBio HiFi covered all genes, except when using whole blood Chemagen. In sample T662828295 using whole blood Chemagen the gene *PDE6G* for sample T662828295 and 14 genes in T668639440 are uncovered. In contrast, we observed that all genes were covered using ONT. Overall, ONT and HiFi (regardless of the one sample that shows low coverage T668639440 whole blood Chemagen) showed the least amount of uncovered medically relevant genes.

To further assess the coverage from a medical perspective, we downloaded SNPs and indels from ClinVar that were reported pathogenic and checked whether the individual sequencing technologies adequately covered these variant locations. We used variants that are not conflicting in reporting their pathogenicity and submitted by multiple clinics (see Methods). This resulted in a set of 10,368 variant sites across the 4641 medically relevant genes.

We calculated the number of uncovered (coverage <1) variants per sample and condition: For the PacBio HiFi sample, on average, we have six variants without coverage for sample T662828295 (only two out of four datasets have uncovered variants) and 24 for T668639440 (three datasets have uncovered variants). ONT covered all variant sites, while Illumina, on average, did not cover six variants across the individual runs for sample T662828295 and seven for T668639440 (Supplementary Data S4).

Next, we analyzed the list of 73 American College of Medical Genetics and Genomics (ACMG) genes, in which mutations are commonly recommended to be reported to patients. These genes span both gene groups defined in this study, including mostly easily accessible genes (68) but also some challenging genes (5). Overall, the genes are well covered by each of the technologies and the normalized average gene coverage is one or more (Supplementary Fig. 10 and Supplementary Data S5). For the five challenging genes, we observed that the normalized average coverage is slightly higher for ONT than for HiFi, indicating a better mapping overall for the ONT data (Supplementary Fig. 11 and Supplementary Data S5). Additionally, for the pilot data T668639440 and T662828295, the normalized mean coverage for accessible genes are similar, HiFi (1.06) and ONT (1.04). Nevertheless, coverage differs in challenging genes, where HiFi is 0.88 and ONT is 1.03. For the benchmark of variants within genes available in the GIAB, the results from HiFi and Illumina are similar compared to ONT. However, the average F-score across these genes is higher for Illumina (93.64%) compared to HiFi (85.24%) and ONT (73.98%) that is due to low F-score for gene *TNNI3* (Supplementary Data S5 and Supplementary Fig. 12). Based on the previous finding, we can conclude that the ACMG list is well covered by all technologies; Likewise, we can call variants with high accuracy using either Illumina or HiFi.

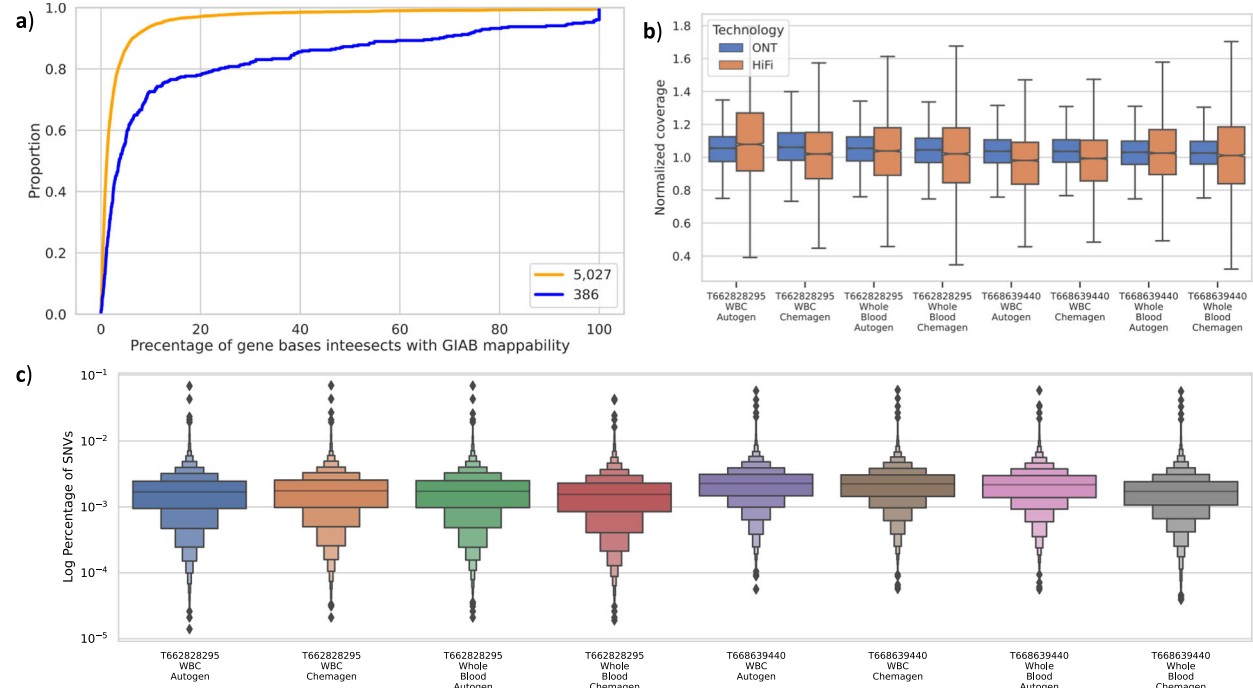

**Fig. 3 | Comparison across medically relevant genes. a** Percentage of gene bases intersecting with hard-to-map regions in yellow is the 5207 genes and in blue 386 medical gene sets. **b** Normalized gene coverage (normalized by sample coverage) for HiFi (yellow) and ONT (blue) for T668639440 and T662828295. **c** Log percentage of SNVs per gene for the 4641 medical genes for samples T668639440 and T662828295.

## Resolution of highly challenging medically relevant genes

We assessed the utility of long-reads specifically within highly complex and repetitive medically relevant genes. In principle, this is where the long-read technology offers its greatest advantages, but it remains to be shown across primary tissues from patient donors. For this, we focused on a gene set from a recent GIAB publication, which proposed 386 genes that were found to be highly challenging for mapping and variant calling[20].

We first assessed the normalized gene coverage (Fig. 4a) between HiFi and ONT. Here, HiFi has a lower median gene coverage than ONT similar to the genome-wide results. Further, normalized coverage distribution is centralized around median 1.05× for ONT compared to 0.97 for HiFi (Supplementary Fig. 13). Additionally, we counted the number of uncovered genes (i.e., genes that do not contain a single read).

For Illumina, nine genes (*CCL3L1, CRYAA, DGCR6, DUX4L1, H19, NAIP, PRODH, SMN1,* and *U2AF1*) are uncovered in sample T668639440 and eight genes (*CCL3L1, CFC1B, DUX4L1, H19, HLA-DRB1, SMN1, TAS2R45,* and *U2AF1*) in sample T662828295. Thus, five genes are uncovered across both samples (Supplementary Data S6). Interestingly, in our analysis, we found that ONT and HiFi did not cover genes *H19* and *U2AF1*. Nevertheless, previous studies found that these genes are incorrectly duplicated in GRCh38, which makes it hard, if not possible, to call variants in these genes[19,50]. However, genes *CCL3L1* and *DUX4L1* are covered, making them only challenging for short reads. Interestingly, the *SMN1* gene differentiates between the ability of long-read to untangle this complex repetitive gene. While HiFi could not support coverage for the gene in all samples, ONT managed to cover it in ~50% bp of sample T662828295. Nonetheless, ONT failed to do the same in sample T668639440.

We next compared the percentage of genes where 50% or more of the gene body lacks coverage (Fig. 4c). In the majority of samples, we saw that the 386 genes group has a higher percentage of gene bodies that are not fully covered. However, for ONT, the percentage of uncovered gene bodies is always lower than HiFi for 386 and 5027

groups alike. Additionally, we saw only that the difference between the two gene sets is in guanine (G) and cytosine (C) GC content percentage (Supplementary Fig. 14). Thus, we conclude that the difference in the percentage of uncovered genes is due to the sample coverage.

We next assessed variant calling ability for long reads, starting with the GIAB sample that has a gold standard benchmark to compare. Specifically, we employed sample NA24385 to rank and characterize 273 genes (70.75% of our 386 gene panel) for which a GIAB benchmark is available. We excluded seven genes that do not report a variant in this benchmark. For each technology, we investigated the top and lowest ten genes ranked by F-score (Supplementary Data S7) and compared these genes across the other technologies. Importantly, the top ten genes that achieved the highest F-scores with Illumina, had the same or better F-score with HiFi. Meanwhile, for these same top ranked genes, ONT had a lower recall for three genes by failing to identify an insertion in each gene (*PIGV* and *MYOT*) and two substitutions in *MYOT*.

For the ten lowest performing genes using HiFi (*CBS, CRYAA, GTF2IRD2, H19, KCNE1, KMT2C, MDK, MUC1, SMN1,* and *TERT*), in six genes (*CBS, CRYAA, GTF2IRD2, KCNE1, MUC1,* and *SMN1*) HiFi still showed a better performance than Illumina. Likewise, ONT achieved a better F-score in these genes than Illumina. Moreover, in *SMN1, KCNE1,* and *CBS* genes, the ONT F-scores are better than HiFi and Illumina. However, in *KMT2C* and *TERT* genes, Illumina outperformed both HiFi and ONT F-scores (Illumina 67.47% and 59.74%), (HiFi 44.76% and 54.32%), and (ONT 32.47% and 47.78%), respectively.

For AoU samples T668639440 and T662828295, because we do not have an established benchmark for these samples, first, we compared the percentage of variants per gene to identify any abnormalities (Fig. 4b). When we consider the distribution of the variants in the 386 genes across datasets (tissue source and extractions), we found that the variant distribution is similar and the tissue source or extraction method did not substantially affect the variant distribution. We then analyzed the concordance of substitutions and indels across the 386 genes, and found the lowest concurrence in introns between the three technologies (approximately 60.47% for exons and 57.67% for introns).

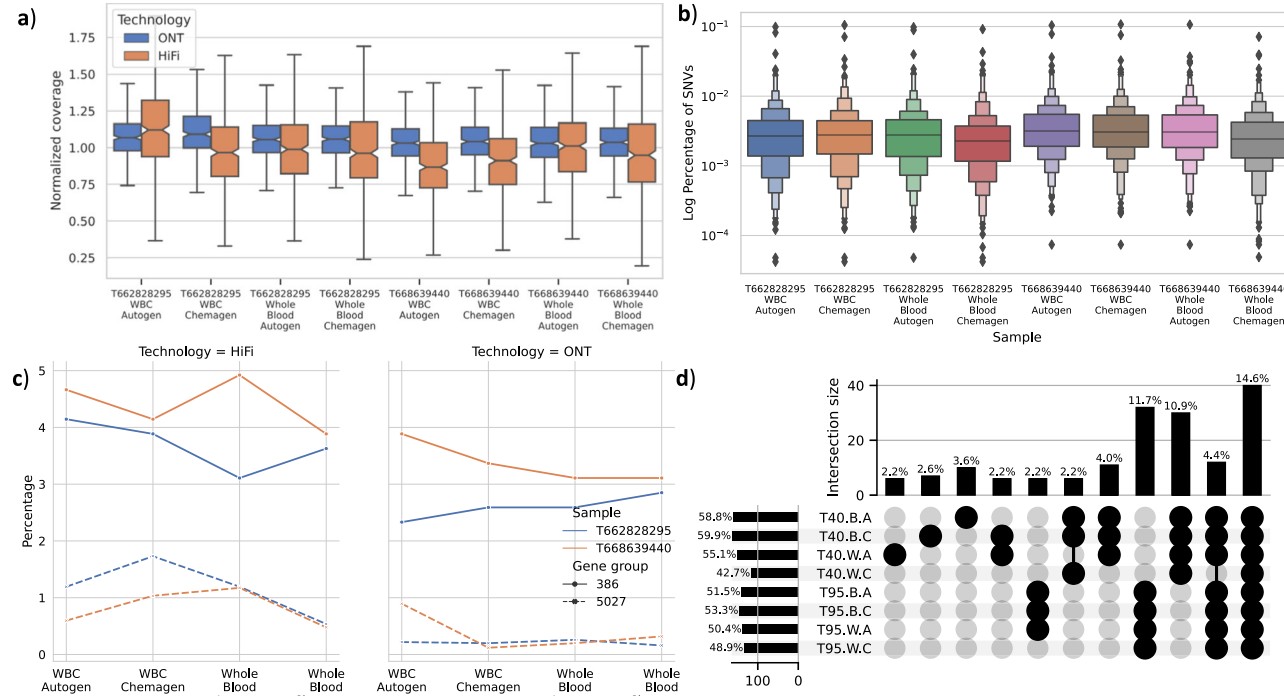

**Fig. 4 | Comparison across 386 challenging medically relevant genes.**
**a** Normalized gene coverage distribution for the 386 genes in different datasets for ONT and HiFi. **b** The percentage of SNVs (substitutions and indels) in the data sets for the 386 genes. **c** Percentage of zero-base coverage per gene, where 50% or more of the gene body is not covered using HiFi and ONT data. **d** SVs breakpoints that intersect with medical genes across samples for different tissue and extraction methods T40 stands for sample T668639440 and T95 for T662828295, W for Whole Blood cells, B for WBC, and finally A for Autogen and C for Chemagen.

For these 386 genes, we observed that the variant calling capability of Illumina seems to be highly impacted, even in exonic regions, compared to the easier set of 5027 medically relevant genes. This is most clear in the overall concordance between short and long reads: For the 5027 genes, we had a high concordance (81.33%) for all three platforms, while for the 386 the concordance drops to 60.74%. The latter is impacted by a reduced concordance of Illumina and showing also higher Illumina-only variants that are likely falsely identified (Supplementary Data S8). Maybe not surprisingly, this trend is amplified for the intronic regions (Supplementary Data S8)

We next investigated the coverage of the ClinVar pathogenic variants that intersect with these genes (see Methods). For HiFi sequenced samples, we achieved on average ~8× coverage for T662828295 and ~9× for the T668639440 sample per variant, both are lower than what the 5027 genes group achieved (9× and 10× respectively) (Supplementary Fig. 15). Likewise, not surprisingly, Illumina achieved the highest average coverage per variant, followed by ONT and HiFi respectively (Supplementary Data S4). Interestingly, when we compare the number of uncovered variants for each sample per technology, we can see a systematic distribution of variants without coverage in Illumina (SD: 1.39) compared to the more variable ONT results (SD: 5.93) (Supplementary Fig. 16 and Supplementary Data S4); However, we could not find a correlation between the gene composition (GC, GT, or AT rich) and the coverage (Supplementary Fig. 17).

In summary, HiFi outperformed the other technologies in both precision and recall (Supplementary Data S7). Furthermore, there are a few genes that are particularly challenging where all technologies missed variants like *H19*, which has one substitution (C/G) (Chr11 at position 1,996,209) that none of the three technologies managed to detect; *KRTAP1-1*, wherein both long-read technologies (HiFi and ONT) did not call any variants, while Illumina called two false-positive substitutions; and *MDK*, where ONT did correctly recall variants, but the other technologies called false-positive variants.

Interestingly, all the genes that ONT detected variants with low F-score were caused by uncalled indels (max three indels) like *FLAD1* and *PIGV* (Supplementary Data S7 shows in more details genes names and F-score and the number of identified variants per technology). Finally, we compared the identified SVs between tissues and extraction in these samples (Fig. 4d) in the medically relevant genes. As we can see, bars 8 and 9 show the unique variants in AoU samples T662828295 and T668639440 supported by different tissue sources and extraction methods; likewise, bar 10 shows the effect of low coverage on sample T668639440 whole blood Chemagen as it lost around 4.4% of SVs that should share among all tissues and extraction methods.

## Discussion

In support of ongoing cutting-edge research in the All of Us project, in this paper, we evaluate the potential use of long-read for All of Us participants. We focus on the advantages and drawbacks of each sequencing technology across different tissues and extraction methods. Likewise, we evaluated methods to call short variants (point mutations, small insertions, and deletions) and structural variants and if using different methods separately or through a merging process will lead to better results. We carried out this analysis across both a set of non-repetitive medically relevant genes and more complex, challenging medically relevant genes including the ACMG list. From these comparisons, we conclude long-reads have widespread value for establishing the most complete and accurate variant calls for All of Us and potentially for many other projects. Moreover, our findings suggest that while previous observations have indicated the reliability of assessing dominant-acting pathogenic mutations from OMIM using short-read whole-genome sequencing (srWGS), our results indicate that these mutations can be discovered with even higher precision and at a reasonable cost.

When comparing the different sequencing technologies, one must consider both the data characteristics (coverage, read length,

error rate, etc.) and the analytical methodologies used. Even though PacBio Sequel II currently produces the lowest coverage runs per sample (usually around 6× to 8×), we could nevertheless use it to accurately identify a majority of SNV and SV calls[51]. In contrast, Illumina-based samples had a much higher coverage (>30× coverage) but suffered from major inherent biases in SV detection, in accordance with previous publications[37]. Thus, simple comparisons of raw sequencing coverage or other simple metrics are not sufficient to evaluate the utility of a sequencing technology. Also, across the two long-read platforms, we found read lengths were not a major distinguishing feature for our variant analysis. For complex variants and extended repetitive regions, it is of course the case that read length is an important factor, which is highlighted by the comparison to the Illumina data sets[6,25]. Nevertheless, when comparing the two long read technologies, ONT average read length (N50) was sometimes larger than that from PacBio, as driven by the tight length distribution for HiFi versus a highly variable distribution from ONT, yet this had marginal to no improvement in variant calling accuracy. Interestingly, while read length is frequently suggested as a dominant factor that may favor ONT, our results demonstrate that the benefits of read length are overshadowed by the higher sequencing accuracy of the HiFi technology. While this is subject to change given sequencing technology updates in chemistry and pore design from ONT and future computational methods, it is still interesting to note that single read accuracy today has a larger impact on variant calling ability. Nevertheless, it is worth highlighting that ONT has recently developed a new method called duplex sequencing. This approach aims to improve read accuracy by sequencing both strands of DNA.

To improve the variant calling accuracy and accessibility across long read cohorts, we release the cloud-based pipeline in Terra (and make the underlying code publicly available on GitHub). Previously, we developed a framework called PRINCESS[51] that phases and calls all types of genomic variants, including SNVs, indels, SVs, and methylation; however, here, we wanted to develop a deliverable pipeline that is cloud accessible. Using the cloud, we were able to optimize variant calling strategies as well as leverage the aspect of running multiple callers, which can be computationally burdensome without elastic computing resources. These pipelines are available for use in the All of Us workbench, along with other instances of Terra, such as the NHGRI AnVIL[52]. They can also be run on institutional servers using a WDL compute engine such as Cromwell (https://cromwell.readthedocs.io/en/stable/) or Miniwdl (https://miniwdl.readthedocs.io/).

Relatedly, we demonstrated that variant calling methods that adopt machine learning or deep learning approaches (e.g., DeepVariant and Clair) are generally performed better than other software that does not (e.g., Longshot). Even more fundamentally, because Longshot does not call indels, it suffers from overall poor recall and poor precision, however, it uses less CPU time than other tools, and no trained model is required. Furthermore, for maximal accuracy, we recommended using a combination of two SNV callers (e.g., Pepper and Clair3). Utilizing both, we developed a merging strategy that yielded high accuracy across both long-read platforms by leveraging the genotypes produced by Clair3, as these were found to be more reliable. Nevertheless, it is worth noting that running both programs also resulted in a large increase in runtime for a marginal improvement in precision (0.01% for DeepVariant and 0.04% for Clair3) but not the overall F-score. Still, our pipeline is now capable of producing high-quality SNV and indel calls across the long-read platforms. The latter was previously considered a major limitation of long-reads, but our work shows it is now possible to capture this important class of variation, as it is now well-established that indels have a major role in many diseases[53]. For SV detection, we confirmed previous reports showing that long-read platforms improved the detection compared to short reads by essentially every metric. As often discussed, this is mainly due to the complexity of SV being larger (50 bp+) than the

Illumina reads itself[6]. We did not compare the ability to identify large CNVs (multiple Mbp) in this study because, to our knowledge, there are currently no specialized tools for long reads that are capable of calling CNVs, despite their association with numerous diseases[54].

We have further showcased the accuracy of long reads across 386 challenging medically relevant genes that are otherwise hard to assess with short reads alone. As previously postulated, we could confirm a substantial improvement in variant calling accuracy and completeness for these genes with long reads[20]. Assessing the true clinical significance of this, however, will require much larger sample sizes, as it is clear they harbor a high degree of polymorphism[55]. Beyond these most difficult genes, we also present the interesting result that long-reads can effectively recover genetic information from a general set of 5000 medically relevant genes. Does the recovery of genetic information from 386 genes justify the use of long reads at scale? In this paper, we also present the interesting result that long-reads can effectively recover genetic information from a general set of 5000 medically relevant genes. In contrast to the 386, these 5000 are not all as challenging yet we that long reads yield measurable value across several metrics. This also holds for the ACMG gene list which is highly important for the medical field.

Thus, the question is what technology is the most appropriate to use at scale within AoU. Based on our results, all three platforms have trade offs. From a production sequencing lab standpoint, Illumina is the only technology demonstrated to scale to one million clinical-grade genomes. Additionally, it is worth mentioning that Illumina continues to develop its analysis platform, DRAGEN, which recently demonstrated significant accuracy in short variant calling. However, our work here as well as other projects demonstrate that long-read technologies are not far behind[31,56–58]. For long reads to advance to the scale of millions of genomes, several major considerations must be addressed including costs, throughput, robustness of software cycles, and predictable/variable yields from sequence components or DNA quality fluctuations. Nevertheless, we believe that the long-read technologies are advancing rapidly in these directions so that AoU and the genomics community at large can now confidently begin such large-scale initiatives.

As we and other recent works show, long-reads have matured significantly over the past 1–2 years, reaching high accuracies for variant identification and also delivering the promises of phasing and methylation (data not shown here). Likewise, in the ACMG[36], which represents a crucial list of genes in the medical field, long-read sequencing demonstrated its efficiency in sequencing those genes and reporting variants more accurately compared to short reads, which is currently the *de facto* approach for analyzing this gene set. Longer term, the question rises if we have entered the age of using long-reads exclusively. We conclude that despite currently scaling and costs considerations, we should continue developing population-scale cohorts sequenced with long reads only. Currently, the primary remaining downside to this approach is a slight reduction in accuracy across small indels, which we anticipate will soon disappear given improvements to the sequencing platforms and their associated computational methods.

Thus, this study shows the strong value of long-reads for simple and complex medically relevant genes and gives clear indications that long-reads are on par with if not better than short-reads. AoU and other population-scale projects should investigate the usage of long-reads at scale and how to utilize and understand the clinical relevance of the so-obtained novel alleles in the setting of larger short-read cohorts.

## Methods

### Ethics

The research held in this manuscript complies with all relevant ethical regulations and is conducted in accordance with the Declaration of Helsinki. Samples were consented with the protocol F-641-5 and

collected from BioChain. The BioChain Institutional Review Board (IRB), chaired by Dr. Zhongdong Liu and comprised of Dr. Vidyodhaya (Vidya) Sundaram as Director of IRB, Dr. Lutong Zhang as Director of IRB, Grace Tian as Director of IRB, and Dr. Ruhong Jiang as Director of IRB, thoroughly reviewed the project application titled "Collection of Leftover Human Specimens that are Not Individually Identifiable." All contingencies have been addressed, leading to the approval of the application by the IRB. All participants provided written informed consent prior to their participation in the study. The data used in this study are subject to controlled access due to privacy concerns. Researchers who wish to access the data can do so by submitting an access request to https://workbench.researchallofus.org/.

### Variant detection coverage analysis

For this analysis, we considered an idealized scenario where reads are error-free and always correctly mapped so that we could isolate the impact of coverage and read length. Following widely used approximations[59], the sequencing coverage is modeled using the Poisson distribution centered on a given overall coverage level. We further assume that a variant must be covered by at least two reads (e.g., min_support = 2), and the coverage will be evenly distributed across maternal and paternal haplotypes following the binomial distribution with $p = 0.5$. We then model the recall of variants as the fraction of genomic positions having coverage with at least 2× coverage. For this, we use the Poisson cumulative density function in R and plot using ggplot. This provides a theoretical upper bound for variant detection for substitutions, deletions, and other small variants where all or nearly all bases of the read align to the reference genome.

We next consider insertion variants where the length of the variant is an appreciable fraction of the read length. This requires additional consideration over deletions or small variants since the insertion sequence will generally not align to the reference genome, at least not at the position of the insertion. In contrast, deletions will have split read alignments, so can be more easily spanned for any length. For this analysis, we consider HiFi-like reads that are all uniformly 20kbp long with a given amount of coverage overall, and at least 2 reads must span the insertion to recall it. For this analysis we again use the Poisson cumulative distribution function in R, but we reduce the effective coverage by the fraction of the insertion length compared to the read length. To derive this, consider N is the total number of reads, G is the length of the genome, R is the length of each read, L is the length of the insertion, and C is coverage. Note by definition, C = NR/G. The probability that a given read spans the insertion is (R-L)/G, so the total number of reads that span the insertion is N(R-L)/G. This can be refactored as (NR/G)(1-L/R) and since (NR/G) defines coverage C this reduces to C(1-L/R). For example if a variant is half as long as the read length (10kbp insertion with 20 kb reads), it will effectively reduce the available coverage in half. This over simplifies the analysis since in practice the read needs to span more than the insertion to be confidently aligned, but establishes an upper bound on recall. As plotted, the recall represents the recall of homozygous variants, although heterozygous variants follow the same distribution as a function of the haplotype specific coverage. The full model and additional simulations are available at: https://github.com/mschatz/coverage_analysis.

### Long read library preparation, QC, and sequencing methods

The following was performed by HudsonAlpha Discovery, a division of Discovery Life Sciences. For all long read assays, stock DNA concentration was measured using the Picogreen assay (Invitrogen), and the DNA size was estimated using the Fragment Analyzer (Agilent). Post DNA QC, approximately 5 µg of stock DNA was sheared to a target size of 20–30 kb on a Megaruptor 3 (Diagenode). The DNA was then purified using 0.45× Ampure XP PB beads with a final elution of 40 µL

Elution Buffer (EB; Qiagen). Post purification, the concentration of the sheared and purified DNA was measured using the Qubit DNA HS assay (Invitrogen), and the DNA size estimation was done using the Fragment Analyzer. Sheared DNA was then size selected using the Pippin HT instrument (Sage Science) with a target range between 15–22 kb. Post size selection, the DNA was then purified using 0.45x Ampure XP PB beads with a final elution of 50 µl EB. Post purification, the concentration of the size-selected and purified DNA was measured using the Qubit DNA HS assay, and the DNA size estimation was done using the Fragment Analyzer. Independent aliquots of the fragmented, purified and size-selected DNA were used in library preparation methods for the Pacific Biosciences Sequel IIe and Oxford Nanopore PromethION platforms For the Pacific Biosciences platform, DNA was taken into circular consensus sequencing (CCS) library prep using the SMRTBell Express Template Prep Kit 2.0 and Enzyme Cleanup Kit 1.0 (PacBio). Each library was barcoded using PacBio Barcoded Overhang Adapters 8 A and 8B (PacBio). Post enzyme cleanup, the libraries were purified 2 times using 1x and 0.6x Ampure XP PB beads with a final elution in 22 µL EB. Post library prep, the concentration of the library DNA was measured using the Qubit DNA HS assay, and the DNA size estimation was done using the Fragment Analyzer. The library concentration and size were entered into SMRTLink (PacBio) for each of the libraries. The target loading concentration was 85 pM with Adaptive loading and no pre-extension with a 30 h movie time. The library annealing, binding, and loading plate worksheet was generated automatically from SMRTLink. Final library binding was done using the Sequel II Binding Kit 2.2 with Sequencing Primer v5. Sequel II DNA Internal Control Complex 1.0 was added to each sample as per manufacturer's recommendation. Sequencing was done on PacBio Sequel IIe running SMRT Link Version 10.1.0.119588.

For the Oxford Nanopore platform, approximately 1 ug of fragmented, purified and size-selected DNA in a volume of 47 ul was used in the SQK-LSK109 library preparation protocol per manufacturer's instructions. This is a ligation-based protocol for the production of libraries compatible with the nanopore sequencing platform. Briefly, the DNA was end-repaired using the NEBNext FFPE DNA Repair Mix and NEBNext Ultra II End Repair/dA-tailing Module reagents in accordance with manufacturer's instructions and placed on ice. The polished DNA was then purified with AMPure XP beads (1:1 vol ratio) and eluted to a final recovered volume of 60 ul. The purified, polished DNA was ligated to sequencing adapters in a 100 ul volume using adapters provided in the LSK109 library preparation kit. Following ligation, the ligated DNA was purified with AMPure beads and using the Oxford Long Fragment Buffer per the manufacturer's direction. Following purification and elution in 51 ul total volume, 1 ul of sample was used to prepare a 1:10 dilution for sample QC using High Sensitivity Qubit and dsDNA Fragment Analysis. Resulting final library yield was 1.2–2.2 ug per sample. Samples were loaded onto the Promethion Flowcells with 20 femto Molar loading. After 24 h, all samples were nuclease washed and reloaded with 20 fM of library. Data was collected on the PromethION platform for a total of 72 h over the sequencing run.

### Pipeline description/method (ONT and HiFi)
**Aligning and coverage.** PacBio data were aligned with pbmm2 (1.4.0) with the parameters (`--preset ccs --strip --sort --unmapped`); MD tags are then added by the `samtools`[60] calmd command (samtools 1.10). ONT data were aligned with minimap2 (`2.17-r941`)[61] with the parameters (`--aYL --MD -x map-ont`). Aligned BAMs of each flow-/SMRT-cell are then merged by sample with `samtools merge` (samtools 1.10). We calculated coverage at the sample level using mosdepth (0.3.1)[62] and per gene average coverage also collected by mosdepth.

Additionally, we calculated per-base coverage using samtools depth (1.15.1) with the parameters (`-a`).

**Calling SNVs**. We used Clair3[43] (v0.1-r6) for each sample, calls are made per chromosome with default parameters, and then merged, and followed with `bcftools sort` (bcftools 1.13)[63].

We additionally applied PEPPER-Margin-DeepVariant[64] for CCS data. For computational efficiency, we parallelized execution per chromosome arm. We then executed the Pepper pipeline[42] (from official docker image kishwars/pepper_deepvariant:r0.4.1) in the following way: `run_pepper_margin_deepvariant call_variant --ccs --phased_output`.

The haplotagged bam output by the previous step is then used by then DeepVariant (from official docker image google/deepvariant:1.2.0) via `/opt/deepvariant/bin/run_deepvariant --model_type=PACBIO --use_hp_information`. The per-chromosome VCFs and gVCFs are then merged by `bcftools concat` (bcftools 1.13), and followed by `bcftools sort` (bcftools 1.13). The single sample VCF is then phased with MarginPhase (from docker image kishwars/pepper_deepvariant:r0.4.1) via `margin phase /opt/margin_dir/params/misc/allParams.phase_vcf.json -M`.

For ONT data, for each sample, the bam is first split in a way that balances the interval sizes. Then Pepper (from official docker image kishwars/pepper_deepvariant:r0.4.1) is run the following way: run_pepper_margin_deepvariant call_variant --gvcf --phased_output --ont. The per-chromosome VCFs phased VCFs, and gVCFs are then merged by `bcftools concat` (bcftools 1.13), and followed by `bcftools sort` (bcftools 1.13).

Finally, for Longshot[41], calls are made on each chromosome with version 0.4.1 and default parameters, and then merged, and followed with `bcftools sort` (bcftools 1.13).

For phasing We used MarginPhase, phasing is done (using docker kishwars/pepper_deepvariant:r0.4.1) via `margin phase /opt/margin_dir/params/misc/allParams.phase_vcf.json -M`.

**Annotating SNVs**. We utilized `SnpEff` version 5.1 to annotate the identified SNVs and indels. The annotation process involved using the default parameters, and we made use of the GRCh38.99 reference from `SnpEff` to annotate the variants. In order to filter out the variants with high impact annotations, we employed `bcftools view` version 1.13 with the option "--include" set to `'ANN~"HIGH"'`. This allowed us to specifically select and extract variants that were annotated as "HIGH" impact according to the annotation information.

**Calling SVs**. For each sample, Sniffles (1.0.12)[12] calls are made per chromosome with custom parameters (`-s 2 -r 1000 -q 20 --num_reads_report —1 --genotype`), and then merged by `bcftools concat` (bcftools 1.13), and followed with `bcftools sort` (bcftools 1.13).

Pbsv: for each sample, PBSV (2.6.0)[65] calls are made per chromosome, and then merged, and followed with `bcftools sort` (bcftools 1.13).

**Annotating SVs**. We utilized `vcfanno` version 0.3.3 to annotate the structural variants (SVs) using the default parameters. For the annotation process, we employed a toml configuration file that specified the GRCh38 annotation bed file. To filter out variants that had no annotations, we utilized `bcftools view` version 1.13 and set the "--exclude" option to `"gene_name = '.'"`. This allowed us to exclude variants that did not have any gene name annotations, ensuring that our analysis focused only on variants with relevant gene information.

**Illumina analysis**. We used Dragen pipeline (v3.4.12) to call variants for Illumina with the default parameters, and we called SVs using Manta[46] (v1.6.0) with the default parameters. Furthermore, we calculated the genome coverage using `mosdepth` (v0.3.2) with four threads, and we set --by to 10 kbp and --mapq to 20. Additionally, we used mosdepth to calculate per gene coverage using bed file of genes coordinate with the --by option and for the normalized gene converge we divided the average gene coverage from the previous step with the average genome coverage of the sample.

**Calculate mappability**. We intersected the gene coordinate for both genes groups (386 and 5027) with the mappability track from the GIAB project (version 2) available at (ftp://ftp-trace.ncbi.nlm.nih.gov/ReferenceSamples/giab/release/genome-stratifications/v2.0/GRCh38/mappability/GRCh38_lowmappabilityall.bed.gz) using `bedtools`[66] (version 2.30.0) `intersect -wo` where -a is the genes and -b is the mappability track, and sum all intersect lengths within the gene divided by gene length to calculator the gene mappability intersect percentage.

**Filter SNVs**. We calculated the number of variants (SNVs and indels) for each tool (Clair3, DeepVariant, and Longshot) and technology (HiFi and ONT) before and after filtering, using AWK and Bcftools view[63]. To select SNVs we used '`bcftools view -H -v snps`', '`bcftools view -i 'strlen(REF)<strlen(ALT)' -H -v indels`' for insertions, and '`bcftools view -i 'strlen(REF)>strlen(ALT)' -H -v indels`' for deletions. Later, we filter the identified variant and choose only pass variants using bcftools `bcftools view -Oz -i 'FILTER="PASS"'`, and count them as mentioned earlier.

We benchmark identified variant (calculate precision, recall, and F-score), for sample "NA24385/HG002" using available truth set from GIAB and RTG[67] tools version 3.12.1 using baseline and bed file supported from GIAB, and for the medical relevant genes (386) we used the specified bed files for these genes from GIAB. For the rest of HapMap samples (HG00514, HG00733, and NA19240) we benchmarked the variant (SNVs and Indels) using the truth set available from Chaisson et al.[44], with RTG tools as mentioned previously. Later, to identify the best combination of technology and tool, we merged the detected SNVs between the two technologies and three tools using Bcftools '`bcftools merge -Oz --threads 2 --force-samples --merge all`' and updated the variant ID with bcftools '`bcftools annotate -Oz --set-id '%CHROM\_%POS\_%REF\_%ALT''`'. Furthermore, we identified the permutation for all technologies and tools in the VCF file using '`bcftools query -f '%ID\t[%GT\t]\n''` and awk, followed by extracting each permutation by ID using bcftools `bcftools view -Oz -i ID=@IDs.txt`'. Additionally, we benchmarked each permutation using RTG and truth set from GIAB for sample "NA24385/HG002" and for the rest of HapMap sample (HG00514, HG00733, and NA19240), first we identified regions in which the truth set did not call variant in it using in house script where SNVs absent in 500 bp or more, later, we removed these regions from our call set and benchmarked them with RTG and appropriate truth set and tools the average of recall, precision, and F-score as an indicator for performance.

For samples T662828295 and T668639440, we agreed to use HiFi data with and merge variants from Clair3 and DeepVariant for each tissue and extraction method using bcftools merge '`bcftools merge --threads 2 --force-samples --merge all`'. To identify which GT to use from the merge, we benchmarked GT for Clair3 and DeepVariant against the truth set using sample NA24385; first, we merged Clair3 calls with DeepVariant as mentioned above and updated their IDs, later, we selected only variants that both tools agreed on using bcftools '`bcftools view -Oz -i 'count(GT = "RR") == 0 && count(GT = "mis")==0'`', then we merged it with the truth set from GIAB NA24385 sample, and selected unmissed variant `bcftools view -i 'count(GT="mis")==0'` and compared the GT between Clair3, DeepVariant, and truth set. We found that Clair3 agrees more with the truth set; thus, we selected it to represent the GT for further analysis. Later, we applied that to samples T662828295 and T668639440 to merge GT before phasing using MarginPhase.

Additionally, we calculated the number of substitutions and indels per gene by intersecting the identified variants with the Bed file for each group of genes (386 and 5027) using bedtools intersect.

**Filter SVs.** We called SVs using Sniffles and pbsv (methods section above). We counted the SVs before filter, later, we filtered SVs based on SV size, where 50 bp is the minimum SV size and considered only pass SVs in case of pbsv using bcftools `bcftools view -Oz -i '(SVLEN >= 50 | SVLEN <= −50 | SVLEN = 0 | SVLEN = 1 | SVLEN=".") & (FILTER = "PASS")'` and we counted them after that. For Sniffles, we further used an in house script to select ~25k SVs based on coverage. Likewise, we selected only deletions and insertions using bcftools `bcftools view -i 'SVTYPE = "DEL" | SVTYPE = "INS"'`, then we merged the identified SVs from technologies (HiFi and ONT) and tools (pbsv and Sniffels) using `SURVIVOR merge` with the following parameters `1000 1 1 0 0 50`. Afterwards, we benchmarked each permutation by filtering them using `bcftools view -i "SUPP_VEC~'X'"`, where X is the permutation we wanted to benchmark, and we used Truvari[68] version 3.0.0 with the following parameters ` --multimatch --passonly -r 2000 --includebed`. For samples, T662828295 and T668639440, we merged the SVs from HiFi call using SURVIVOR merge[69] with the following parameters `1000 2 1 0 0 50`.

**Source of pathogenic variant and extraction.** We downloaded the ClinVar variants (SNVs and indels) from "https://ftp.ncbi.nlm.nih.gov/pub/clinvar/vcf_GRCh38/clinvar_20220320.vcf.gz". We selected only pathogenic variants with a provided criterion, multiple submitters, and no conflict using bcftools `bcftools view -i 'CLNSIG = "Pathogenic" & CLNREVSTAT = "criteria_provided\,_multiple_submitters\,_no_conflicts" & CHROM! = "X" & CHROM! = "MT"'` (hereafter referred to as pathogenic variants). Further, we renamed the chromosomes to match our data using `bcftools annotate -Oz --rename-chrs`. Later, we intersected the variant with both gene groups (386 and 5027) bed files using `bedtools intersect`.

To calculate coverage, we transform the pathogenic variant VCF file to bed using bedops[70] vcf2bed, for deletions `vcf2bed --deletions`, insertions `vcf2bed --insertions`, SNVs `vcf2bed --snvs` and lastly we merged them all using bedops --everything, then intersect it with per-base coverage file for each gene group using bedtools intersect `bedtools intersect -wo` and `-a` is the pathogenic variant and `-b` is per-base coverage.

### Reporting summary
Further information on research design is available in the Nature Portfolio Reporting Summary linked to this article.

## Data availability
The genome data generated in this study have been deposited in the All of Us workbench database under https://www.researchallofus.org/data-tools/workbench. The genomic data are available under restricted access for human subject data, access can be obtained by following the instructions under All of Us workbench.

## Code availability
The pipeline code can be found at https://doi.org/10.5281/zenodo.10419695.

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

## Acknowledgements

This manuscript was supported by the Baylor/JHU award (OT2 OD002751, 3OT2OD002751). The All of Us Research Program is supported by the National Institutes of Health, Office of the Director: Regional Medical Centers: 1 OT2 OD026549; 1 OT2 OD026554; 1 OT2 OD026557; 1 OT2 OD026556; 1 OT2 OD026550; 1 OT2 OD 026552; 1 OT2 OD026553; 1 OT2 OD026548; 1 OT2 OD026551; 1 OT2 OD026555; IAA #: AOD 16037;

Federally Qualified Health Centers: HHSN 263201600085U; Data and Research Center: 5 U2C OD023196; Biobank: 1 U24 OD023121; The Participant Center: U24 OD023176; Participant Technology Systems Center: 1 U24 OD023163; Communications and Engagement: 3 OT2 OD023205; 3 OT2 OD023206; and Community Partners: 1 OT2 OD025277; 3 OT2 OD025315; 1 OT2 OD025337; 1 OT2 OD025276. In addition, the All of Us Research Program would not be possible without the partnership of its participants. E.E. is an investigator of the Howard Hughes Medical Institute.

## Author contributions

M.M., E.E., S.L., M.T., M.S., K.G., and F.S. designed the study. M.M., K.G., Y.H., P.A., W.W., and F.S. analyzed the data. F.S., E.E., S.L., M.T., and M.S. provided critical feedback and oversaw the project. N.P., S.H., R.H., and A.P. performed the study analysis, including sequencing the samples. All authors contributed to writing the manuscript.

## Competing interests

F.S. received support from Illumina, PacBio, Oxford Nanopore Technologies, and Genentech. S.H. received support from HudsonAlpha Institute for Biotechnology and Owens Cross Roads. E.E. is a scientific advisory board (SAB) member of Variant Bio, Inc. The remaining authors declare no competing interests.
