## [Peer Review File · Nature Communications]

Utility of long-read sequencing for All of UsREVIEWER COMMENTS

Reviewer #1 (Remarks to the Author):

The paper "Utility of long-read sequencing for All of Us" by Mahmoud et al. investigates the use of long-read sequencing for variant calling in human genomes.

Overall I find the study relevant and appreciate the approach of focusing on medically relevant genes that are either "simple" or "challenging" to sequence using short-read platforms. However, although the aim is highly relevant, too many fundamental method errors prevent an in-depth review.

In line 208, the authors state, "we assessed the ability of the technologies for the identification of SNVs and indels across these samples using state-of-the-art methods." and line 485, "In support of ongoing cutting-edge research in the All of Us project." Furthermore, in the abstract, the authors conclude: "Our results show that HiFi reads produced the most accurate results for both small and large variants.". This conclusion has already been propagated by PacBio, which quotes it widely on social media.

However, I do not understand how the researchers reached this conclusion. Some of the authors even participated in another study that states that ONT outperforms Illumina and PacBio: <https://www.biorxiv.org/content/10.1101/2023.01.12.523790v1>. It seems like this study was "frozen" 2-3 years ago, and a lot has happened since.

I have three major points that need to be addressed before a proper review can be conducted:

1) Technology. The authors used what I suspect is ONT R9 data. The materials and methods do not state it but refer to a very old Nanopore sequencing kit (SQK-LSK109). The R9 flowcells are only in legacy support, and the data quality from R10.4 with kit14 + the latest basecaller is dramatically better than R9 data. The authors also need to supply information on the Nanopore basecalling software, which is just as essential as stating the (also missing) pore version. I recommend re-running the samples with the latest chemistry and basecalling software.

2) Software. My area of expertise is not human-related variant calling, but out of interest, I randomly checked if the latest Illumina DRAGEN software was used. The authors report the use of the DRAGEN pipeline v3.4.12. This version was released in November 2019, and the current version is v4.0.3 from July 2022. I have not used this pipeline myself, but performance would usually increase significantly over 2.5 years spanning 18 new releases, with several major updates included. Therefore, I recommend checking all software used in the study and ensuring that essential parts are using the latest version of the software available.

3) Approach. I do not understand the reasoning behind the difference in sequencing coverage allocation from each technology. For example, 4-6 PacBio flowcells is allocated pr. sample, but only 1-2 Nanopore flowcells pr. sample. The variant calling is highly dependent on both error-rate and coverage. Hence to be a fair comparison between technologies, it would need to be compared how much data you can get for x-amount of money. This would also be a more realistic use case for users who implement these studies in practice. In my lab, for the same amount of money, I could buy 1 PacBioB sequell IIe flowcell for every 3 PromethION R10.4 flowcells. Hence, even though HiFi data is of higher quality, the 5-10x difference in coverage for the same amount of money would likely have a dramatic advantage in favor of ONT.

However, prices change over time. Hence, in benchmark studies, I highly prefer coverage-dependent evaluations. As the authors write, coverage is essential in understanding all metrics reported here. Hence, I recommend to subset each dataset in intervals from 5-60X and calculating the statistics over each dataset. That would enable a more robust coverage-independent comparison where fx. 5X PacBio

performance could be compared with 30X Nanopore performance.

I highly favor a simple comparison on one sample with a coverage gradient, compared to the current situation with multiple samples with multiple levels of coverage - it is simply impossible to make a meaningful comparison across these datasets.

Reviewer #2 (Remarks to the Author):

The authors evaluate and contrast the use of different sequencing technologies for detecting variation in human genomes. In particular, they compare the Illumina platform with two long-read technologies. The first part of the paper focuses on SNV and SV calling comparisons genome-wide while the remainder of the paper is focused on medically relevant genes including genes that are difficult to sequence using short reads (Wagner 2022). Although the paper is quite comprehensive, the conclusions of the paper are not very substantive and it is not clear what the novel findings are. For example, the abstract states "Our analysis revealed substantial differences in the ability of these technologies to accurately sequence complex medically relevant genes, particularly in terms of gene coverage and pathogenic variant identification." The Wenger et al. 2019 paper and recent papers have reached similar conclusions.

Figure 1 presents an idealized recall of variants as a function of coverage, however, no results from real data that correspond to variant detection as a function of coverage were shown. In addition, to evaluate the impact of sequence coverage on variant detection, one would expect that the samples would be sequenced at high coverage. However, the long-read sequencing coverage for many of the samples, particularly the AoU samples was low. For some of the samples, the HiFi coverage was only 5-10x.

Some of the explanations for discordant or unexpected findings are not well justified. For example, line 239-240:

" This might be because of different challenges in variant representation or comparison, as well as less training data specifically in these samples". There are several such examples.

No de novo assembly based methods were used for variant calling. One would expect to see something on this front especially if the goal is to compare the power of short-read with long-read sequencing technologies.

Many of the analyses are based on four DNA samples from the All of Us project. It is not clear what is the advantage of analyzing DNA samples from this project as opposed to publicly available samples from 1000 Genomes or the GIAB consortium. I am not sure that cell lines or even the sequence data from this project is publicly available.

Reviewer #3 (Remarks to the Author):

Dr. Mahmoud and co-authors present the evaluation of long-read sequencing for participants in the All of Us project. Extensive evaluation of SNV, indel, and SV detection algorithms was performed. The authors conclude that (cost equal, perhaps), among the long-read technologies, the HiFi sequencing achieves greater accuracy than the ONT sequencing. Other metrics, such as the high quality SNV calls by short reads are in accordance with prior expectation.

The manuscript is an important study to guide decision making for a resource-intensive project such as AoU. The technical implementation is excellent for SV analysis, however there are some shortcomings addressed later.

Major 1. The primary comment is an elephant in the room: a study evaluating technologies, particularly long-read sequencing will suffer from being obsolete not only upon publication, but perhaps even before the study is completed. This is acknowledged in the discussion, where it is stated that "this is subject to change given sequencing technology updates from ONT" as well as the introduction that mention technology developments from both companies. However, the moving target of technology evaluation should be embraced from the onset rather than mentioned in the discussion. I was hoping to see that the methods were cloud enabled so that new versions of each technology could be evaluated and disseminated real time to the research community. Fortunately, this is the case.

The cloud based analysis should be mentioned at the beginning, and it should be made clear how to run new data through the analysis pipeline for the genomes that have assembly based callsets.

Major 2. A large emphasis is placed on concordance between callsets, without focusing on the nature of the calls. The number of loss of function variants in high pLI (low oe) genes should be calculated per technology for SNV and frameshift indel variants. The analysis should be in-depth. The difference in the number of SNV calls between ONT and pacbio is staggering - over 350k variants! It seems strange to have such similar F-scores between the methods.

Major 3. Similar to SNV calls, an emphasis is placed on number of SVs and concordance rather than impact. It has been shown that much of the long-read variant calls are in repetitive DNA, and short-read variant callsets discover impactful variation.

To quote one of the authors of this study:

As we note above, srWGS captures virtually all high-quality deletions derived from lrWGS assembly in the regions of the genome that encompass over 95% of currently annotated coding sequence in genes with existing evidence for dominant-acting pathogenic mutations from OMIM. (Zhao 2021).

The analysis from Figure 3A and B from this paper would help readers interpret the impact that long-read variant discovery has on interpretable SVs. If the authors disagree with that quote, rationale or data should be presented as to why.

Minor comments.

Minor 1. Do the methods detect any copy number variation in genes? This would be a high-impact variant missed by gene count.

Minor 2. A phrase is repeated lines 551-555.

Minor 3. Typo line 576.

We would like to thank you for your valuable feedback, insightful comments, and constructive suggestions, which have contributed to improving the quality of our manuscript. We carefully considered each of the reviewers' comments and have made the necessary revisions accordingly, including a few new supplemental figures. Over the past weeks, we have integrated the requested additional analysis and comments into the manuscript. We highlighted these changes in blue in the revised manuscript. Furthermore, we replied to the comments of the reviewers in detail, which are also highlighted in blue below.

Reviewer #1:

The paper "Utility of long-read sequencing for All of Us" by Mahmoud et al. investigates the use of long-read sequencing for variant calling in human genomes.

Overall I find the study relevant and appreciate the approach of focusing on medically relevant genes that are either "simple" or "challenging" to sequence using short-read platforms. However, although the aim is highly relevant, too many fundamental method errors prevent an in-depth review.

In line 208, the authors state, "we assessed the ability of the technologies for the identification of SNVs and indels across these samples using state-of-the-art methods." and line 485, "In support of ongoing cutting-edge research in the All of Us project." Furthermore, in the abstract, the authors conclude: "Our results show that HiFi reads produced the most accurate results for both small and large variants.". This conclusion has already been propagated by PacBio, which quotes it widely on social media.

However, I do not understand how the researchers reached this conclusion. Some of the authors even participated in another study that states that ONT outperforms Illumina and PacBio: <https://www.biorxiv.org/content/10.1101/2023.01.12.523790v1>. It seems like this study was "frozen" 2-3 years ago, and a lot has happened since.

I have three major points that need to be addressed before a proper review can be conducted:

1) Technology. The authors used what I suspect is ONT R9 data. The materials and methods do not state it but refer to a very old Nanopore sequencing kit (SQK-LSK109). The R9 flowcells are only in legacy support, and the data quality from R10.4 with kit14 + the latest basecaller is dramatically better than R9 data. The authors also need to supply information on the Nanopore basecalling software, which is just as essential as stating the (also missing) pore version. I recommend re-running the samples with the latest chemistry and basecalling software.

We would like to thank the reviewer for his comments, the ONT that was used to conduct this analysis is R9, and we used kit SQK-LSK109, and we have added this information to the main text. R10.4 with kit14 was officially released on 16th Nov 2022 (<https://community.nanoporetech.com/posts/rapid-sequencing-kit-v14>) only around 2 weeks

before this paper was submitted. Thus, it was effectively impossible to sequence all samples with R10 kit14. We had eluded to the issue of the rapidly evolving ONT chemistry in the discussion section.

We have revised this section of the methods, ensuring that it clearly specifies the kit we used: "For the Oxford Nanopore platform, approximately 1ug of fragmented, purified, and size-selected DNA in a volume of 47ul was used in the SQK-LSK109 library preparation protocol, following the manufacturer's instructions."

With regard to the highlighted paper by Kolmogorov et al. that paper actually does not include any Pacbio data. Kolmogorov et al only compare Oxford Nanopore to matching Illumina data and don't focus on medically important genes, which is in contrast to our paper and is one of the main objectives of All of Us. Both papers are using R9 flow cells and the main difference might be the extraction. Also Kolmogorov et al mainly focus on the substitution for the comparison and ignoring indels.

2) Software. My area of expertise is not human-related variant calling, but out of interest, I randomly checked if the latest Illumina DRAGEN software was used. The authors report the use of the DRAGEN pipeline v3.4.12. This version was released in November 2019, and the current version is v4.0.3 from July 2022. I have not used this pipeline myself, but performance would usually increase significantly over 2.5 years spanning 18 new releases, with several major updates included. Therefore, I recommend checking all software used in the study and ensuring that essential parts are using the latest version of the software available.

It is true, we used DRAGEN version v3.4.12, which is not the current release. This is because this version is the only version that AoU has been approved to use by the US Food and Drug Administration (FDA), which is one of the governing bodies of AoU (See <https://developer.illumina.com/dragen/dragen-popgen>). Therefore, to be comparable to the core AoU program itself, we kept this version for our analysis. We do envision there are improvements with the newest version, although it is hard to validate new versions as DRAGEN releases a new version every 6–8 months. We have added a statement in the discussion to address this.

Furthermore, we have included an additional sentence to explicitly clarify the reason for using this specific version: "DRAGEN version 3.4.12, which was selected for this project as it is the approved version by the Food and Drug Administration."

3) Approach. I do not understand the reasoning behind the difference in sequencing coverage allocation from each technology. For example, 4-6 PacBio flowcells is allocated pr. sample, but only 1-2 Nanopore flowcells pr. sample.

The variant calling is highly dependent on both error-rate and coverage. Hence to be a fair comparison between technologies, it would need to be compared how much data you can get for x-amount of money.

This would also be a more realistic use case for users who implement these studies in practice. In my lab, for the same amount of money, I could buy 1 PacBioB sequell IIe flowcell for every 3

PromethION R10.4 flowcells. Hence, even though HiFi data is of higher quality, the 5-10x difference in coverage for the same amount of money would likely have a dramatic advantage in favor of ONT.

In our manuscript, our first objective was to achieve high coverage of both ONT and HiFi (>20x) technologies in order to benchmark their ability to accurately identify variants (SNVs, indels, and SVs), irrespective of the yield per flow cell. To this end, we utilized a range of flow cells per technology to achieve the desired level of coverage. This allowed us to assess the performance of both technologies across a wide range of data yields, providing valuable insights into their respective capabilities for variant detection. This is particularly relevant with the recent release of the Revio which has substantially dropped the cost of sequencing with PacBio, and is now installed at several of the AoU sequencing centers.

In the second part of our study, we focused on comparing the results obtained from each technology by using one to two flow cells for each. Specifically, we compared the normalized gene coverage by sample coverage to determine if each technology was able to cover complex genes effectively, regardless of the initial coverage. This approach allowed us to make a more accurate comparison of the performance of each technology.

However, prices change over time. Hence, in benchmark studies, I highly prefer coverage-dependent evaluations. As the authors write, coverage is essential in understanding all metrics reported here. Hence, I recommend to subset each dataset in intervals from 5-60X and calculating the statistics over each dataset. That would enable a more robust coverage-independent comparison where fx. 5X PacBio performance could be compared with 30X Nanopore performance.

I highly favor a simple comparison on one sample with a coverage gradient, compared to the current situation with multiple samples with multiple levels of coverage - it is simply impossible to make a meaningful comparison across these datasets.

We acknowledge your point and fully agree with your request for a straightforward comparison. However, it is important to consider that sequencers can exhibit varied performance when analyzing different types of samples, such as cell lines versus DNA samples derived from blood. As a result, conducting a comparison across multiple samples becomes necessary. In our study, we have performed the experiment using coverage ranging from 5x to 25x, and we have provided a detailed report on this in the main text. Additionally, we have included a new supplemental figure (also displayed below) that displays the F-scores for each coverage set across both platforms for HG002, along with the corresponding SV f1-score in CMRGs (GIAB benchmark).

We have included this paragraph under the “Optimizing variant detection in cell lines” section on page 7 to emphasize the significance of the newly added experiment:

“Furthermore, we conducted a titrated coverage comparison for sample HG002, focusing on the two long-read technologies, PacBio and ONT. We evaluated the SV detection F1-score for this

comparison, and the results can be found in Supplementary Figure 18. Overall, the trends follow the idealized results presented in Figure 1, with a sharp increase in accuracy between 5x to 15x coverage and align with other ongoing research⁴⁷. We also observed that below 8x coverage, ONT demonstrated a higher F1 score for SVs compared to PacBio HiFi. However, beyond 8x coverage, there was a significant improvement in the F1-score for PacBio HiFi, exceeding 90%, while the F1-score for ONT remained around approximately 87%.”

Reviewer #2:

The authors evaluate and contrast the use of different sequencing technologies for detecting variation in human genomes.

In particular, they compare the Illumina platform with two long-read technologies. The first part of the paper focuses on SNV and SV calling comparisons genome-wide while the remainder of the paper is focused on medically relevant genes including genes that are difficult to sequence using short reads (Wagner 2022).

1. Although the paper is quite comprehensive, the conclusions of the paper are not very substantive and it is not clear what the novel findings are. For example, the abstract states "Our analysis revealed substantial differences in the ability of these technologies to accurately sequence complex medically relevant genes, particularly in terms of gene coverage and pathogenic variant identification." The Wenger et al. 2019 paper and recent papers have reached similar conclusions.

We would like to thank the reviewer for their comment. In our manuscript, we compared a set of 4,641 genes that are typically sequenced using Illumina technology. Our results show that long reads from PacBio or ONT perform equally or even better than Illumina in this set of genes. Furthermore, we specifically examined a set of medically complex relevant genes (386 which were published in 2022) and found that using long reads in such cases is superior to Illumina due to limitations such as read length and mappability. Therefore, we suggest that using long-read sequencing for medically challenging genes will not sacrifice any performance, and in fact, could provide better results. This applies not only to medically complex relevant genes but also to other medically relevant genes for which, at this moment, Illumina is the de facto choice for medical research.

2. Figure 1 presents an idealized recall of variants as a function of coverage, however, no results from real data that correspond to variant detection as a function of coverage were shown. In addition, to evaluate the impact of sequence coverage on variant detection, one would expect that the samples would be sequenced at high coverage. However, the long-read sequencing coverage for many of the samples, particularly the AoU samples was low. For some of the samples, the HiFi coverage was only 5-10x.

To address this comment, we have now also included a subsampling benchmark from GIAB to measure the F-score (arithmetic mean of precision and recall). We included these results as a new supplementary figure and summarized the results in the main text. What is great to see is that Supplementary Figure 18 (also displayed below) has a high similarity with the idealized results presented in Figure 1. This comparison is a bit skewed, as Figure 1 only plots recall probabilities without considering precision or sequence-specific biases (e.g. GA repeats for HiFi reads, for example, are known to have reduced coverage). Nevertheless, the similarities are striking across the coverage points, illustrating how coverage is one of the main factors in detecting variations, especially for 5x to 15x coverage. Our measurements with ONT are below both HiFi and idealized recall for higher coverage ranges, which we attribute to a low number of SV in the GIAB benchmark for CMRG and the higher per-read error rate. Finally, we also highlight very recent work by Harvey et al. demonstrating similar trends for variant recall and precision using long read sequencing (<https://www.biorxiv.org/content/10.1101/2023.05.04.539448v1>)

3. Some of the explanations for discordant or unexpected findings are not well justified. For example, line 239-240: " This might be because of different challenges in variant representation or comparison, as well as less training data specifically in these samples". There are several such examples.

We thank the reviewer for highlighting these, and we have worked on revising them and giving more insights. To address this, we added this section to the manuscript under the "Optimizing variant detection in cell lines" section on page 7:

"The difference in the accuracy of variant calling between HG002 and other samples is attributed to the truth set bed regions for HG002 being filtered to exclude repetitiveness or polymorphic complexity regions, as mentioned in Wagner *et al.*³⁸ and Li *et al.* studies⁴⁵. Additionally, the variants in this sample have been extensively validated, and their accuracy is well-established"

4. No de novo assembly based methods were used for variant calling. One would expect to see something on this front especially if the goal is to compare the power of short-read with long-read sequencing technologies.

In our manuscript, our primary focus was to have a direct comparison across the technologies. This is currently possible for a mapping-based comparison, as assembler software quality varies significantly across these technologies. Pacbio HiFi currently leads in assembly quality, especially software developed by the HPRC and other projects. While for ONT the currently proposed assemblers generally do not even provide phasing of the haplotypes (e.g. Shasta).

Thus, we excluded these benchmarks for now also because we used assembly-derived variant benchmarks such as GIAB and wanted to avoid a bias.

5. Many of the analyses are based on four DNA samples from the All of Us project. It is not clear what is the advantage of analyzing DNA samples from this project as opposed to publicly available samples from 1000 Genomes or the GIAB consortium.

We have opted to use AoU donor DNA samples instead of solely relying on 1000 Genomes or GIAB cell lines. This decision is based on the understanding that sequencers, particularly ONT, can exhibit varied performance depending on the input material. By including DNA samples, we can gain insights into the behavior of these samples across the entire program. Additionally, the AllOfUs samples have been selected as clinical controls, further emphasizing their importance in the study.

Furthermore, during the early stages of the long-read pilot, the consortium recognized that certain sample extraction methods yielded better results than others. However, the impact of these differences on variant calling itself was not adequately elucidated at the time. In our study, we aim to address this knowledge gap by thoroughly documenting and describing the specific impacts of these variations in extraction methods on variant calling.

6. I am not sure that cell lines or even the sequence data from this project is publicly available.

All the data used in our manuscript is publicly available through the research workbench provided by the AoU initiative (<https://www.researchallofus.org/data-tools/workbench/>).

For additional information regarding data availability and accessing the research bench, please refer to the following link: <https://allofus.nih.gov/get-involved/opportunities-researchers>. This resource provides detailed information and guidance for researchers interested in accessing the data and utilizing the research bench.

Reviewer #3:

Dr. Mahmoud and co-authors present the evaluation of long-read sequencing for participants in the All of Us project. Extensive evaluation of SNV, indel, and SV detection algorithms was performed. The authors conclude that (cost equal, perhaps), among the long-read technologies, the HiFi sequencing achieves greater accuracy than the ONT sequencing. Other metrics, such as the high quality SNV calls by short reads are in accordance with prior expectation.

The manuscript is an important study to guide decision making for a resource-intensive project such as AoU. The technical implementation is excellent for SV analysis, however there are some shortcomings addressed later.

1. Major 1. The primary comment is an elephant in the room: a study evaluating technologies, particularly long-read sequencing will suffer from being obsolete not only upon publication, but perhaps even before the study is completed. This is acknowledged in the discussion, where it is stated that “this is subject to change given sequencing technology updates from ONT” as well as the introduction that mention technology developments from both companies. However, the moving target of technology evaluation should be embraced from the onset rather than mentioned in the discussion.

Thank you for your comments. We agree this is a challenge, but nevertheless, it is important to measure the current performance as the project is underway. We expanded the introduction to address this point, page 3:

“The rapid advancements in long-read sequencing platforms necessitate continuous monitoring of their performance with respect to multiple variant types and regions of the genome.”

2. I was hoping to see that the methods were cloud enabled so that new versions of each technology could be evaluated and disseminated real time to the research community. Fortunately, this is the case. The cloud based analysis should be mentioned at the beginning, and it should be made clear how to run new data through the analysis pipeline for the genomes that have assembly based callsets.

We appreciate the reviewer’s comments, and thank them for their efforts in reviewing our manuscript. We acknowledge their concerns regarding the pipeline and the ever-evolving landscape of technologies. We are pleased to say our pipeline is now frozen and is being used for all long-read data sets within the All of Us program, and it is likely to be adopted by other consortia as well. We want to clarify based on the reviewer's last sentences that this pipeline is developed to utilize raw long reads (Pacbio or ONT) and provides mapping-based variant calling. We did the comparison to assembly-based variant call sets (HG00514, HG00733, and NA19240), which are not automated, since the purpose of the pipeline is to analyze hundreds to thousands of long reads data sets over the cloud.

- Major 2. A large emphasis is placed on concordance between callsets, without focusing on the nature of the calls. The number of loss of function variants in high pLI (low oe) genes should be calculated per technology for SNV and frameshift indel variants. The analysis should be in-depth. The difference in the number of SNV calls between ONT and pacbio is staggering - over 350k variants! It seems strange to have such similar F-scores between the methods.

This is indeed a very interesting comment, and we have now worked this in into the analysis We added this new section to the results:

“We utilized SnpEff version 5.1 to annotate the merged variants obtained from the three technologies, as outlined in the methods section. A total of 10,939,305 variants were processed and annotated, resulting in 41,050,827 effects being assigned to these variants. Among them, we identified 7,622 variants (0.02%) across 24 data sets (three technologies were used to analyze two samples, each of which was obtained using two different extraction methods and sourced from two different tissues, resulting in a total of 24 datasets.) with high-impact annotations, including stop-gained or frameshift variants, which were of particular interest for further analysis. When comparing the high-impact variants identified by ONT, HiFi, and Illumina, we found that the percentage of high-impact variants exclusively detected by Illumina was the same as those detected by both HiFi and ONT (0.04%), as well as HiFi and Illumina. However, considering the high-impact variants detected by either ONT or HiFi alone, this percentage increased to 0.05%. Additionally, when evaluating all long-read technologies collectively, the percentage of high-impact variants identified rose to 0.09%, as depicted in Supplementary Figure 19.”

In response to the second point, “the difference in the number of SNV calls between ONT and PacBio is staggering:

The precision of SNV calls varied significantly when utilizing the DeepVariant and Clair3 tools for HiFi and ONT data. Specifically, the precision for HiFi data was notably higher, with DeepVariant achieving a precision of 99.92% and Clair3 achieving a precision of 99.89%. In contrast, the precision for ONT data was lower, with DeepVariant achieving a precision of 98.31% and Clair3 achieving a precision of 97.28%. These findings are visualized in Supplement Figure 4.

4. Major 3. Similar to SNV calls, an emphasis is placed on number of SVs and concordance rather than impact. It has been shown that much of the long-read variant calls are in repetitive DNA, and short-read variant callsets discover impactful variation.

We would like to thank the reviewer for calling our attention to this. We added this new section to describe the impact of SVs on gene bodies. We also note repeat expansions/contractions and other intergenic variations are known to be associated with several major diseases so this analysis is certainly undercounting the potential clinical significance of long read SVs:

“Additionally, we utilized vcfanno to annotate the structural variants (SVs) that we identified, as described in the methods section. We calculated the percentage of SVs that overlap genes (entire gene body) and presented these findings in Supplementary Figure 19. This might be inflated as larger inversions for example might not have a direct impact.

Our analysis revealed that the percentage of SVs overlapping genes was lower in Illumina sequencing (~43%) compared to HiFi and ONT sequencing (~54%), as well as all SVs identified by long-read sequencing (~52%).”

To quote one of the authors of this study:

As we note above, srWGS captures virtually all high-quality deletions derived from lrWGS assembly in the regions of the genome that encompass over 95% of currently annotated coding sequence in genes with existing evidence for dominant-acting pathogenic mutations from OMIM. (Zhao 2021). The analysis from Figure 3A and B from this paper would help readers interpret the impact that long-read variant discovery has on interpretable SVs. If the authors disagree with that quote, rationale or data should be presented as to why.

There is ample supporting literature that reinforces our claim regarding the limitations of short reads, particularly in their alignment to repeat-rich regions, segmental duplications, tandem repeats, and low-complexity regions enriched for GC or AT content (PMC7509619, PMC8979283, PMC10167679, PMC9706577, and PMC9117392). Furthermore, it is worth noting that over 1000

protein-coding genes, which have medical relevance, are associated with these regions (PMC9186530). Finally, this comment was specifically about high quality deletions, meaning insertions, inversions, and other SV types can still play a major role.

Minor comments.

Minor 1. Do the methods detect any copy number variation in genes? This would be a high-impact variant missed by gene count.

There currently aren't robust methods for long read-based large CNV detection available that are well tested. The SV callers capture a large range of variants, but might fail if a breakpoint of the SV is inside a non-resolved region. Thus, chromosome arm amplification or deletions would be missed. Fortunately, we have presumably healthy individuals here without abnormal karyotypes. We added to the discussion a comment that we have not assessed CNV calling, which presents an opportunity for future work.

Minor 2. A phrase is repeated lines 551-555.

We would like to thank the reviewer for pointing out the duplication; we removed the redundant section.

Minor 3. Typo line 576.

We have corrected the typo

REVIEWERS' COMMENTS

Reviewer #2 (Remarks to the Author):

My overall opinion about the manuscript has not changed in light of the revised manuscript and the authors' response to my comments. I think that the manuscript can be substantially improved in terms of thoroughness and completeness of the analysis. However, the revised manuscript did not attempt to do so.

For example, in response to comment #2, the authors subsample long-read sequence data from 5-25x coverage. However, as also mentioned by reviewer #1, sequencing and sub-sampling from higher depth of coverage (60x) is important to fully understand the impact of sequence coverage.

Similarly, the response to #5 is not satisfactory. It is still not clear to me whether this study's findings are applicable to only the All of Us project or in general to long read sequencing projects for humans. The AllofUs DNA samples used in this study are not openly available for sequencing limiting the use of such samples for future comparisons (e.g. using newer version of sequencing technology) by others.

Reviewer #3 (Remarks to the Author):

The authors have largely responded to my comments. The following new text needs rewording, but not re-review.

When comparing the high-impact variants identified by ONT, HiFi, and Illumina, we found that the percentage of high-impact variants exclusively detected by Illumina was the same as those detected by both HiFi and ONT (0.04%), as well as HiFi and Illumina.

==> as well as the union of HiFi and Illumina?

The vcfanno results should probably exclude inversions, partially because the inversion calls seem to still be difficult to make (in larger inversions), and because as stated, it may inflate the results.

No re-review needed.

I still feel the response to the comment below was insufficient, and should be addressed in the text. In light of the Cell paper, one could view the results of this paper that lrWGS adds a few hundred additional genes missed by srWGS, and if the Cell paper results hold, relatively few new pathogenic deletions will be found. Pathogenic duplications, as noted in the manuscript, are not assessed.

It does not need re-review. A comment along the lines the following would address the concern.

"While it has been previously observed that dominant-acting pathogenic mutations from OMIM may be reliably assessed using srWGS, these results indicate the mutations may be discovered with higher precision and reasonable cost."

Previous comment:

As we note above, srWGS captures virtually all high-quality deletions derived from lrWGS assembly in the regions of the genome that encompass over 95% of currently annotated coding sequence in genes with existing evidence for dominant-acting pathogenic mutations from OMIM. (Zhao 2021). The analysis from Figure 3A and B from this paper would help readers interpret the impact that long-read variant discovery has on interpretable SVs. If the authors disagree with that quote, rationale or data should be presented as to why.

There is ample supporting literature that reinforces our claim regarding the limitations of short

reads, particularly in their alignment to repeat-rich regions, segmental duplications, tandem repeats, and low-complexity regions enriched for GC or AT content (PMC7509619, PMC8979283, PMC10167679, PMC9706577, and PMC9117392). Furthermore, it is worth noting that over 1000 protein-coding genes, which have medical relevance, are associated with these regions (PMC9186530). Finally, this comment was specifically about high quality deletions, meaning insertions, inversions, and other SV types can still play a major role.

Dear Editor and reviewers,

We would like to thank you for your valuable feedback, insightful comments, and constructive suggestions, which have contributed to improving the quality of our manuscript. We carefully considered each of the reviewers' comments and have made the necessary revisions accordingly. We highlighted these changes in blue in the revised manuscript. Furthermore, we replied to the comments of the reviewers in detail, which are also highlighted in blue below.

As discussed over emails all the data is now available in the researcher workbench (<https://www.researchallofus.org/data-tools/workbench/>). In addition we are investigating a possible deposit of the raw data to ANVIL which should further promote the usage of the data.

As requested here is a brief summary statement:

Using All of Us pilot data, researchers compared short- and long-read performance across medically relevant genes and showcased the utility of long reads to improve variant detection and phasing in easy and hard to resolve medically relevant genes.

We would like to highlight three twitter accounts:

@MedhatMahmoud_

@sedlazeck

@BCM_HGSC

Reviewer #2 (Remarks to the Author):

My overall opinion about the manuscript has not changed in light of the revised manuscript and the authors' response to my comments. I think that the manuscript can be substantially improved in terms of thoroughness and completeness of the analysis. However, the revised manuscript did not attempt to do so.

For example, in response to comment #2, the authors subsample long-read sequence data from 5-25x coverage. However, as also mentioned by reviewer #1, sequencing and sub-sampling from higher depth of coverage (60x) is important to fully understand the impact of sequence coverage.

We would like to thank the reviewer for their comments. However, we would like to clarify that the samples we benchmarked, HG002, exhibit a maximum coverage of 30x, as illustrated in Figure 2. Regrettably, benchmarking exceeding this threshold is not feasible in our specific example.

Similarly, the response to #5 is not satisfactory. It is still not clear to me whether this study's findings are applicable to only the All of Us project or in general to long read sequencing projects for humans. The AllofUs DNA samples used in this study are not openly available for sequencing limiting the use of such samples for future comparisons (e.g. using newer version of sequencing technology) by others.

We would like to extend our appreciation to the reviewer for their insightful comment. However, it's essential to emphasize that the primary focus of this paper lies in assessing the utility of long-read sequencing in cohort analysis, with a specific emphasis on medically relevant genes. Every large population study will have its protocol for DNA extractions and other parameters of analysis and library preparation. Nevertheless, AoU represents the current state of the art on how to achieve this at scale with multiple sample collection sites.

Reviewer #3 (Remarks to the Author):

The authors have largely responded to my comments. The following new text needs rewording, but not re-review.

When comparing the high-impact variants identified by ONT, HiFi, and Illumina, we found that the percentage of high-impact variants exclusively detected by Illumina was the same as those detected by both HiFi and ONT (0.04%), as well as HiFi and Illumina.

==> as well as the union of HiFi and Illumina?

We reworded the sentence to be:

“When we compared the high-impact variants identified by ONT, HiFi, and Illumina, we discovered that the percentage of high-impact variants detected exclusively by Illumina was identical to those detected by both HiFi and ONT, which was just 0.04%. The same percentage was also observed for high-impact variants detected by the union of HiFi and Illumina.”

The vcfanno results should probably exclude inversions, partially because the inversion calls seem to still be difficult to make (in larger inversions), and because as stated, it may inflate the results.

No re-review needed.

We have detected 310 inversions among the 19,872 structural variants (SVs), constituting 1.55% of the dataset. To ensure accuracy, we have removed these inversions, recalculated the percentage, and compared the revised results to the previous findings. We also updated Figure 19, which has been included as a supplementary figure.

In addition, we have included the following paragraph to integrate with the previously added text in our manuscript:

“Furthermore, when we filtered out inversions and recalculated the percentages of SVs overlapping genes, we noted a decrease of less than one percentage, as illustrated in Supplementary Figure 19.”

I still feel the response to the comment below was insufficient, and should be addressed in the text. In light of the Cell paper, one could view the results of this paper that lrWGS adds a few hundred additional genes missed by srWGS, and if the Cell paper results hold, relatively few new pathogenic deletions will be found. Pathogenic duplications, as noted in the manuscript, are not assessed.

It does not need re-review. A comment along the lines the following would address the concern.

"While it has been previously observed that dominant-acting pathogenic mutations from OMIM may be reliably assessed using srWGS, these results indicate the mutations may be discovered with higher precision and reasonable cost."

Previous comment:

As we note above, srWGS captures virtually all high-quality deletions derived from lrWGS assembly in the regions of the genome that encompass over 95% of currently annotated coding sequence in genes with existing evidence for dominant-acting pathogenic mutations from OMIM. (Zhao 2021). The analysis from Figure 3A and B from this paper would help readers

interpret the impact that long-read variant discovery has on interpretable SVs. If the authors disagree with that quote, rationale or data should be presented as to why.

There is ample supporting literature that reinforces our claim regarding the limitations of short reads, particularly in their alignment to repeat-rich regions, segmental duplications, tandem repeats, and low-complexity regions enriched for GC or AT content (PMC7509619, PMC8979283, PMC10167679, PMC9706577, and PMC9117392). Furthermore, it is worth noting that over 1000

protein-coding genes, which have medical relevance, are associated with these regions (PMC9186530). Finally, this comment was specifically about high quality deletions, meaning insertions, inversions, and other SV types can still play a major role.

We would like to thank the reviewer for their feedback, and we have added the suggested paragraph in the discussion section of the manuscript.

“Moreover, our findings suggest that while previous observations have indicated the reliability of assessing dominant-acting pathogenic mutations from OMIM using short-read whole-genome sequencing (srWGS), our results indicate that these mutations can be discovered with even higher precision and at a reasonable cost.”